# Microglia-Mediated Neuroinflammation Through Phosphatidylinositol 3-Kinase Signaling Causes Cognitive Dysfunction

**DOI:** 10.3390/ijms26157212

**Published:** 2025-07-25

**Authors:** Mohammad Nazmul Hasan Maziz, Srikumar Chakravarthi, Thidar Aung, Phone Myint Htoo, Wana Hla Shwe, Sergey Gupalo, Manglesh Waran Udayah, Hardev Singh, Mohammed Shahjahan Kabir, Rajesh Thangarajan, Maheedhar Kodali

**Affiliations:** 1School of Medicine, Perdana University, Damansara Heights, Kuala Lumpur 50490, Malaysia; mohammad.nazmul@perdanauniversity.edu.my (M.N.H.M.); manglesh@perdanauniversity.edu.my (M.W.U.);; 2Faculty of Medicine, Nursing and Health Sciences, SEGi University, Kota Damansara Campus, Petaling Jaya 47810, Selangor, Malaysia; srikumarc@segi.edu.my; 3Department of Pathology, Manipal University College Malaysia, Persimpangan Batu Hampar, Bukit Baru, Melaka 75150, Malaysia; 4International Medical School, Management and Science University, Shah Alam 40100, Selangor, Malaysia; phone_myint@msu.edu.my (P.M.H.);; 5Faculty of Medicine and Health Sciences, UCSI University, Springhill, Port Dickson 71010, Negeri Sembilan, Malaysia; 6Saint James School of Medicine Anguilla, 6X75+64G A-1, Albert Lake Dr, The Quarter 2640, Anguilla; 7Department of Cell Biology and Genetics, Texas A&M Health Science Center, College of Medicine, College Station, TX 77843, USA

**Keywords:** microglia, PI3K signaling, neuroinflammation, cognitive dysfunction, cytokine release, synaptic plasticity, Akt-mTOR pathway, neurodegeneration

## Abstract

Microglia, as the immune guardians of the central nervous system (CNS), have the ability to maintain neural homeostasis, respond to environmental changes, and remodel the synaptic landscape. However, persistent microglial activation can lead to chronic neuroinflammation, which can alter neuronal signaling pathways, resulting in accelerated cognitive decline. Phosphoinositol 3-kinase (PI3K) has emerged as a critical driver, connecting inflammation to neurodegeneration, serving as the nexus of numerous intracellular processes that govern microglial activation. This review focuses on the relationship between PI3K signaling and microglial activation, which might lead to cognitive impairment, inflammation, or even neurodegeneration. The review delves into the components of the PI3K signaling cascade, isoforms, and receptors of PI3K, as well as the downstream effects of PI3K signaling, including its effectors such as protein kinase B (Akt) and mammalian target of rapamycin (mTOR) and the negative regulator phosphatase and tensin homolog (PTEN). Experiments have shown that the overproduction of certain cytokines, coupled with abnormal oxidative stress, is a consequence of poor PI3K regulation, resulting in excessive synapse pruning and, consequently, impacting learning and memory functions. The review also highlights the implications of autonomously activated microglia exhibiting M1/M2 polarization driven by PI3K on hippocampal, cortical, and subcortical circuits. Conclusions from behavioral studies, electrophysiology, and neuroimaging linking cognitive performance and PI3K activity were evaluated, along with new approaches to therapy using selective inhibitors or gene editing. The review concludes by highlighting important knowledge gaps, including the specific effects of different isoforms, the risks associated with long-term pathway modulation, and the limitations of translational potential, underscoring the crucial role of PI3K in mitigating cognitive impairment driven by neuroinflammation.

## 1. Introduction

Cognitive impairment is associated with a wide range of deficits in memory, task execution, focus, and learning, commonly linked to neurodegenerative conditions including Alzheimer’s disease (AD), Parkinson’s disease (PD), multiple sclerosis (MS), traumatic brain injury (TBI), and several psychological disorders. Recent evidence suggests that these cognitive deficits are not solely the result of neuronal degeneration but often stem from complex neuroimmune processes involving non-neuronal cells like microglia [1,2,3].

Microglia serve as fundamental immune sentinels of the brain and play a role in essential synaptic pruning, neuronal maintenance, and the resolution of short-lived inflammatory responses. When microglia become persistently activated or dysregulated, they undergo phenotypic shifts that lead to chronic neuroinflammation, which in turn causes excessive and pathological rates of neuronal destruction and loss of neuroplasticity, all of which are closely linked to cognitive dysfunction [4,5]. The phosphoinositide 3-kinase (PI3K) signaling pathway is crucial to the process of microglial activation, particularly PI3K-1 among the three subclasses [6]. PI3Ks are lipid kinases that catalyze the conversion of phosphatidylinositol 4,5-bisphosphate (PIP2) into phosphatidylinositol 3,4,5-trisphosphate (PIP3). This reaction initiates downstream signaling cascades involving protein kinase B (Akt), the mammalian target of rapamycin (mTOR), and nuclear factor kappa B (NF-κB). These pathways can be activated in response to stimuli such as Toll-like receptors (TLRs), cytokines, or damage-associated molecular patterns (DAMPs); Figure 1 [7,8,9].

Such a downstream process is crucial for triggering inflammatory responses, which also determines the microglial polarization state. The M1 and M2 polarization model, which describes microglia as pro-inflammatory and anti-inflammatory, respectively, demonstrates the response of these cells to PI3K signaling, even though it is overly simplistic [10,11,12]. Microglial PI3K-Akt-mTOR pathway signaling is associated with the overproduction of the pro-inflammatory cytokines interleukin-1β (IL-1β), tumor necrosis factor-alpha (TNF-α), and interleukin-6 (IL-6), amplifying neuronal excitotoxicity, synaptic degeneration, and increasing blood–brain barrier (BBB) permeability [13,14,15].

The PI3Kγ and PI3Kδ pathways in microglial PI3K are expressed and functionally specialized separately for neuroimmune responses (Table 1). PI3Kγ is associated with pro-inflammatory and chemotactic signaling, whereas PI3Kδ participates in the regulation of cysteine proteases and phagocytosis, both of which are crucial for maintaining healthy synapses and preserving cognitive functions. Furthermore, the heightened levels of pro-inflammatory cytokines disrupt synaptic plasticity in the hippocampus and impede long-term potentiation (LTP), which is considered the neurophysiological basis of memory [16,17], leading to cognitive dysfunction. Additionally, altered PI3K activity leads to deficient microglia-initiated synaptic pruning, resulting in either the removal of synapses or abnormal connectivity [18]. Furthermore, oxidative stress and mitochondrial health are impacted by microglial PI3K dysfunction, which further reinforces neurotoxic feedback loops [19].

Multiple preclinical studies have demonstrated that the genetic and pharmacological regulation of PI3K in microglia could reverse cognitive impairment. Cognitive features of neurodegenerative diseases were found to be less severe in PI3Kγ-deficient mice, where microglial activation was attenuated alongside improved spatial memory performance [20]. Rodent models also exhibited restorative effects on hippocampal learning coupled with suppressed cytokine release from microglial cells following PI3K blockade with inhibitors LC294002 and wortmannin [21,22,23]. This evidence highlights a strong therapeutic strategy targeting PI3K signaling pathways aimed at modulating microglial function to combat cognitive decline. Moreover, human studies have collaborated on the translational relevance. Increased PI3K activity is evident in the postmortem brains of patients diagnosed with AD and other neurodegenerative disorders in association with CD68, a marker of activated microglia [24]. Also, neuroimaging studies using positron emission tomography (PET) tracers for microglial activation, such as translocator protein (TSPO) ligands, demonstrated spatial overlap with cognitive deficits coupled with stronger PI3K pathway activation of the given region [25,26].

Despite the compelling evidence, there are still some gaps that need further consideration. The specific contributions of each isoform, PI3Kα, PI3Kβ, PI3Kδ, and PI3Kγ, to microglial function remain unclear (see Table 1). While PI3Kγ is largely known for its role in immune system regulation, PI3Kδ appears to control chemotaxis as well as the process of phagocytosis [27,28,29]. Furthermore, any potential adverse impacts of long-term PI3K suppression are concerning given its roles in insulin signaling pathways, cell survival functions, and autophagy [30]. PI3Kγ and δ isoforms are predominant in the immune system, especially in microglia, mediating the inflammatory and chemotactic activities [31], which might be associated with cognitive decline. Microglia surrounding amyloid plaques in APP/PS1 and other transgenic AD models (3xTg-AD) show overexpression of PI3Kγ and δ isoforms, along with elevated IL-1β and TNF-α production, mitochondrial dysfunction, and synaptic demolition [32,33,34]. Moreover, these changes are responsive to PI3K inhibition, where cytokine production is reduced, and spatial memory performance is enhanced.

**Table 1 ijms-26-07212-t001:** PI3K isoforms and their microglial expression patterns.

PI3K Isoform	Class	Expression in Microglia	Functional Relevance	References
**PI3Kα**	Class IA	Low to moderate; implicated in cell survival	Contributes to metabolic control; less understood in neuroinflammation	[7]
**PI3Kβ**	Class IA	Moderate; modulates actin cytoskeleton and vesicular trafficking	Involved in phagocytic cup formation and motility	[27,28]
**PI3Kγ**	Class IB	High; key role in immune responses and chemotaxis	Strongly linked to neuroinflammatory response and cognitive impairment	[27,28]
**PI3Kδ**	Class IA	Moderate to high; regulates inflammation and phagocytosis	Modulates cytokine release and autoimmunity pathways	[27,28]

With the increasing complexities of microglial subtypes and region-specific cognitive impairments, a systems biology approach is essential to unravel the chronological and spatial aspects of PI3K-mediated neuroinflammation. Consequently, this review integrates perspectives from molecular biology, immunology, behavioral neuroscience, and pharmacology to provide a cohesive insight into the role of PI3K-activated microglia in cognitive decline. In the subsequent sections, we will define and describe the constituent elements of PI3K signaling in microglia, evaluate the models of experimentation designed to study this pathway, translate findings from human studies, and devise innovative treatment plans aimed at restoring cognition through modulation of the inflammatory nexus.

## 2. PI3K Activation in Neuroinflammatory Conditions

The PI3K signaling pathway is important for cell survival, proliferation, metabolism, and motility. For microglia, PI3K acts as a crucial intracellular transducer that associates extracellular danger signals with intracellular transcriptional programs. In microglia, PI3K acts as a crucial microglial transducer that links extracellular danger signals to intracellular transcriptional programs. Classical TLRs, interleukin receptors, and even some purinergic receptors trigger microglial activation by recruiting PI3K complexes to the plasma membrane. Activation appears at the plasma membrane. Receptor/ligand interaction at Toll-like receptors, interleukin receptors, and even purinergic receptors leads to the recruitment of PI3K complexes to the plasma membrane [35,36,37]. PI3K phosphorylates PIP2 on position two, turning it to PIP3 with Akt and mTOR, and NF-kB at its downstream—not forgetting the role of membrane PI3K in inflammation, phagocytosis, and oxidative metabolism [38,39,40].

Cognitive impairment due to inflammation has also been modeled through the systemic application of lipopolysaccharide (LPS), which is a microglial-activating bacterial endotoxin. Object recognition deficits and memory impairments observed in LPS-treated mice were associated with increased PI3K-Akt-NF-κB signaling, alongside some depressive behaviors. These deficits were reversible with PI3K inhibitors LY294002 and wortmannin, suggesting that behavioral function depends on this signaling pathway [41]. Traumatic brain injury (TBI) represents another example in which signaling through the PI3K pathway is strongly implicated. The mechanical injury to the brain tissues translates to an inflammatory damage response, and microglial cells are recruited and activated. There is a strong association between the activation of the PI3K-Akt signaling pathway and microglial cell proliferation, as well as cognitive decline with the passage of time after injury [21]. Blocking the pathway activated post-injury prevents not only the hyperactivation of microglia but also promotes the recovery of spatial learning as assessed by the Morris water maze. Hypoperfusion sets a model of further hypoperfusion, chronic, to demonstrate advanced vascular contributions to cognitive impairment, leading to the disruption of white matter and memory function. Reactive astrocytes and microglia showed a marked increase in PI3K/mTOR signaling. Work has been conducted, demonstrating the reversibility of cognitive dysfunction and gliosis through the inhibition of mTOR, a downstream target of PI3K [42].

Additional understanding arises from models of β-amyloid oligomer infusion, where hippocampal activation of PI3K is linked with lower dendritic spine density, increased phagocytosis, and even anxiety-like behaviors. Disorders targeting PI3K show partial restoration of marked synaptic disruption as well as normalization of associated behaviors [23]. Like senescence-accelerated mouse prone 8 (SAMP8) mice, these accelerated aging models exhibit progressive microglial PI3K activity, prominent with memory decline, indicating some vulnerability axis dependent on aging [43]. Their overview is provided in Table 2, which encapsulates neuroinflammation prompted by PI3K activity relative to cognitive outcomes spanning diverse systems. Each study highlights the role of PI3K in performance linkage to cognition, as, in all cases, cognitive gain is observed after modulation through genetic knockout or pharmacological blockade.

These forms of investigation are supported with research that includes human subjects. The postmortem analysis performed on the brains of patients with AD has repeatedly demonstrated increased expression of various isoforms of PI3K within microglia that are located at the site of amyloid plaques and dystrophic neurites [24]. Also, the PET imaging conducted with the use of TSPO ligands of the second generation has shown overactive microglia in areas with early Alzheimer’s disease continuum, including the hippocampus and posterior cingulate cortex. More importantly, these areas show the altered expression of genes and proteins of the PI3K pathway, which suggests a blend between inflammation and destroyed circuits within the neural systems [26].

In addition, some PI3K genes have been connected to an increased risk of neurodegenerative diseases through genome-wide association studies (GWAS). For instance, the genes PIK3CD and PIK3C3 that code for the PI3K isoform are linked to higher chances of AD and Parkinson’s disease, respectively [44]. From these findings, it can be inferred that one might have a genetic risk factor for cognitive dysfunction that is mediated by inflammation if they suffer from high PI3K signaling.

Further aging changes the picture in the following way. With regard to the brain, the condition is termed “primed” microglia, which means that with age, inflammation is partially pre-emptively mitigated due to epigenetic changes and the decline of mitochondria, which makes cells more sensitive to inflammatory signals. This condition is a result of a basal increase in PI3K signaling, likely as a form of compensatory mechanism that paradoxically lowers the threshold for harmful activation. Because of this hypersensitivity, a very mild stimulus could lead to exaggerated inflammation and synaptic damage in elderly people [15].

The entire history of microglial development from anatomical curiosities to immunocompetent regulators of cognition demonstrates the growing importance of glia in basic as well as applied neuroscience. It also shows how important the PI3K pathway is in these intersecting phases, as it provides a mechanistic link between activation states and behavior. Understanding the diversity of microglial context-responsive changes in activation as well as isoform-specific signaling calls for systems-level exploration in future investigations. Only through integrating the molecular, cellular, circuitry, and behavioral frameworks can we disentangle the role of PI3K-driven microglial activity in cognitive impairment and disease progression.

## 3. The PI3K Signaling Pathway: A Molecular Primer

### 3.1. Structural Architecture and Functional Classification of PI3K Enzymes

The PI3Ks are a conserved family of lipid kinases that manage several key biological functions like cellular growth, metabolism, apoptosis, immune system activation, and synaptic plasticity. In the scope of the central nervous system, PI3Ks are being recognized as critical modulators of microglial activation and functions. PI3Ks respond to a wide range of extracellular cues, including trophic factors, neurotransmitters, and DAMPs/PAMPs [28]. PI3K responds integratively to various physiological receptor families and thereby acts as a major bioinformation processor in the microglial response to physiologic situations and neuropathological insults.

PI3Ks are grouped into three classes, Class I, II, and III, due to their sequence homology, regulatory subunits, and substrate preference proteins [27]. Of these, Class I PI3Ks are most relevant to neuroinflammation. They are heterodimeric enzymes composed of a p110 catalytic subunit (p110α, p110β, p110δ, or p110γ) and a regulatory subunit (such as p85 or p101). Class IA enzymes (PI3Kα, β, δ) are activated by receptor tyrosine kinases (RTKs) while Class IB (PI3Kγ) is primarily activated by G protein-coupled receptors (GPCRs) [45]. These isoforms show differential expression in cell types; notably, PI3Kγ and PI3Kδ are preferentially expressed in microglia and are critically involved in innate immune responses [9].

The primary catalytic activity of PI3Ks is to phosphorylate the third position of the inositol ring of PIP2, yielding PIP3. This conversion recruits proteins containing PH domains, such as Akt, which translocate to the plasma membrane and set off a signal transduction cascade that governs cellular decisions concerning metabolism, survival, cytokine production, and motility (Akt) [8].

Research has shown that different microglial function outcomes may result due to differential isoform activation. For instance, PI3Kγ is known to be important for chemotaxis and the release of inflammatory cytokines, while PI3Kδ is associated with microglial phagocytosis and dictating survival [46]. The complexity of immune signaling within the brain is captured by the precise coordination of these isoforms within spatially defined microglial groups. Aberrant PI3K signaling is increasingly recognized as a driver of neuroinflammatory pathology, particularly in the context of chronic or dysregulated activation [47].

### 3.2. Downstream Signaling Pathways: Akt, mTOR, NF-κB, and Feedback Loops

Akt (or protein kinase B) is subject to membrane translocation, where it is phosphorylated at threonine 308 by PDK1 and at serine 473 by mTORC2, achieving full activation. As a convergence point for multiple signaling networks, Akt alters cellular functions such as glucose utilization, protein synthesis, autophagy, apoptosis, and immune response modulation [48]. In microglia, activated Akt facilitates the transcription of inflammatory mediators and increases the uptake of glucose, which is used to support the glycolytic shift in reactive glial phenotypes [49].

A major downstream effector of Akt is the mammalian target of rapamycin (mTOR), a serine/threonine kinase that distinguishes two complexes: mTORC1 and mTORC2. In microglia, the activation of mTORC1 is both autophagy-inhibiting and pro-inflammatory gene expression-driving, contributing to the establishment of a chronic inflammatory phenotype [50]. Elevated mTORC1 activity has been noted in microglial cells during the early stages of AD, after suffering from a traumatic brain injury, or in the early stages of Parkinson’s disease, correlating with higher concentrations of IL-1β and TNF-α and reduced phagocytic activity [51]. Nuclear factor kappa B (NF-κB) is yet another critical downstream target of PI3K-Akt signaling. Following stimulation, PI3K-Akt signaling drives IκB kinase (IKK), which is responsible for the phosphorylation and degradation of IκBα, thus liberating NF-κB to go to the nucleus, where it can then translocate. NF-κB also drives the transcription of various inflammatory genes such as IL-6, C-X-C motif chemokine ligand 1 (CXCL1), chemokine C-C motif ligand 2 (CCL2), and many others [24]. In LPS-induced neuroinflammation models, blocking PI3K or Akt causes NF-κB to lose its ability to come into the nucleus, thus reducing the inflammation-inducing cytokines [52].

Activity within the PI3K-Akt-mTOR signaling pathway is further modified by feedback mechanisms. One such mechanism is the Tuberous Sclerosis (TSC) 1/2 complex, which provides negative regulation for mTORC1, acting as a feedback closure point. Akt phosphorylation, which occurs simultaneously with the inhibition of TSC2, frees mTORC1 from TSC2′s blockade, permitting a positive feedback loop that perpetuates inflammation. Moreover, the chronic activation of mTORC1 causes a feedback inhibition of the insulin receptor substrate-1 (IRS-1), disrupting the balance of glucose homeostasis, which in turn may worsen metabolic dysfunctions in microglial cells [53]. These feedback loops highlight the difficulties for pharmacological approaches that seek to target the pathway while avoiding off-target effects. The importance of these converging signaling pathways is showcased in transcriptomic datasets of human brains affected by AD where microglial clusters located around the plaques and enriched with theories exhibit coordinated upregulation of PI3K-Akt-mTOR and NF-κB targets. This expression profile is confined to degenerative hotspots, indicating that the activation of this signaling axis is context-dependent, as shown by single-nucleus RNA-sequencing [54].

### 3.3. Regulation by Phosphatases: Phosphatase and Tensin Homolog (PTEN), Src Homology 2-Containing Inositol Phosphatase-1 (SHIP1), and Balancing Microglial Activation

The balance in the PI3K pathway is maintained by lipid phosphatases that inhibit the spread of signals within the cells and mitigate the risks of overactivation. PTEN is one of the most characterized lipid phosphatases. It counteracts the actions of PI3K by dephosphorylating PIP3 to PIP2, which halts the recruitment of Akt and PDK1. In microglia, PTEN helps maintain quiescent immune and metabolic homeostasis. However, deletion of PTEN genetically in rodents has been shown to cause hyperactivated microglia, elevate ROS production, and induce neurotoxicity, which suggests that PTEN serves the very important function of limiting neuroinflammation [55].

Another crucial regulatory phosphatase is SHIP1, which hydrolyzes PIP3 into PI (3,4) P2, which is a signaling intermediate with selective and different binding preferences. SHIP1 works along with PTEN and is critical for the survival and migration of immune cells. Microglia in SHIP1-deficient mice tend to have increased expression of inflammatory cytokines, decreased β-amyloid clearance, and worsened cognitive decline in transgenic AD models [56]. Additionally, it has been shown that loss of SHIP1 disrupts the regulation of microglial mitochondrial dynamics, which causes persistent fragmentation of mitochondria and reduced adenosine triphosphate (ATP) production during inflammation [10].

The interaction of PI3K, PTEN, and SHIP1 forms a rheostat-like mechanism that calibrates the level of microglial activation. This balance shifts in response to hypoxia, ATP, glutamate, or DAMPs released due to cellular injury. In the absence of disease, results from PH domains indicate that PI3K activation is temporary and is terminated by active phosphatases. However, in the context of chronic neurodegeneration or enduring infections, this feedback loop is often hijacked, leading to pathologically sustained microglial polarization [15]. Research derived from the transcriptomic and proteomic analyses of the brains of aged mice suggests a loss of facial map PTEN and SHIP1 expression with age, which also stems from heightened basal PI3K signaling and a primed state of microglia. These data imply that the age-associated decline of phosphatases may drive brains toward such inflammatory responses that are exacerbated and sustained over long periods [57].

### 3.4. Isoform-Specific Signaling and Translational Implications for Cognitive Dysfunction

The selective microglial functions of PI3Ks have drawn remarkable attention as they pertain to both the pathophysiology and treatment [58]. While PI3Kα and PI3Kβ have broader expressions in neural and endothelial tissues, PI3Kγ and PI3Kδ are more restricted to the leukocyte lineage, which includes microglia and macrophages that infiltrate. PI3Kγ is known to have a role in the LPS-induced production of cytokines, chemokine receptor signaling, and cell migration. The use of PI3Kγ inhibitors such as IPI-549 and AS605240 can dampen microglial reactivity and preserve cognitive functions in rodent models with traumatic brain injuries and neurodegenerative disorders [59]. PI3Kδ is involved in the microglial control of synaptic integrity. Genetic and pharmacologic inhibition of PI3Kδ—notably with the FDA-approved inhibitor idelalisib—has been shown to preserve synaptic integrity and improve cognitive outcomes in rodent models of chronic neuroinflammation and neurodegeneration [60]. Idelalisib, a selective inhibitor of PI3Kδ approved for treating hematologic malignancies, was shown in preclinical studies to cross the BBB, where it modulates glial activation, suggesting possible central nervous system (CNS) repurposing [61]. Nonetheless, chronic inhibition of PI3Kδ might dampen immunosurveillance, especially in aged populations, which raises concerns for clinical translation.

In addition, the selective inhibition of isoforms affects both energy metabolism and the dynamics of mitochondria in microglia. The blockage of PI3Kγ has been linked with increased mitochondrial biogenesis and the recovery of oxidative phosphorylation, as opposed to PI3Kδ inhibition, which prefers glycolytic compensation [62]. These metabolic influences impact microglial endurance within sustained inflammation, thereby determining the effectiveness of therapeutic interventions aimed at the modulation of immune responses. Microglial activation is not uniform across brain areas. As the hippocampus and prefrontal cortex are the most susceptible to cognitive decline, these microglial regions bear greater expression of PI3Kγ and its upstream modulators Akt2 and mTORC1 under inflammatory conditions. This form of spatial diversity has also been observed in single-cell RNA sequencing and immunofluorescence studies in rodent and human post-mortem tissues [63].

From a therapeutic perspective, the dual inhibition of PI3K and mTOR integratively targets both inflammatory signaling and protein synthesis, presenting a new avenue for focused intervention. Inhibitors of PI3K/mTOR, such as dactolisib (BEZ235) and GNE-477, exemplify this class by effectively dampening microglial cytokine expression and amyloid deposition in AD models but are hampered by toxicity and immunosuppression [64]. It is still a challenge to target the neuroinflammatory process effectively while retaining key immune surveillance roles of the microglia. Taken together, these observations emphasize the intricacy of PI3K signaling in microglia. Therapeutic approaches aimed at this signaling pathway need to consider the effects of isoform-specific and temporal dynamics, compensatory signaling, regional microglial heterogeneity, and even homing microglial responses. The precision in how we modulate this axis to treat cognitive dysfunction is likely to expand with the increasing application of multi-omics and machine learning to model signaling interplay and their impact on therapy outcomes.

## 4. Microglial PI3K Activation in Response to CNS Insults

### 4.1. Infection and Inflammatory Stimuli: Acute Activation of the PI3K Axis

Microglia are primary immune guardians of the central nervous system (CNS) that respond swiftly to invading microbes and inflammation. Among diverse immune stimuli, LPS, a structural component of the outer membrane of Gram-negative bacteria, has become one of the most employed neuroinflammatory agents in vivo. LPS administration either peripherally or via intracerebroventricular routes binds to and activates TLR4 on microglial membranes, leading to the rapid activation of downstream kinases, in particular, PI3K isoforms like PI3Kγ. Following LPS treatment, robust phosphorylation of Akt and NF-κB is detectable in microglia together with the marked secretion of pro-inflammatory cytokines such as IL-1β, TNF-α, and IL-6 [65,66].

This process enhances neuronal excitotoxicity, contributes to the storm cytokine breakdown of the BBB, and subsequently leads to the synaptic loss, more vividly in hippocampal circuits that aid in memory formation. Object recognition, spatial learning, and social preference are consistently behaviorally assessed; these metrics are shown to be deficient after LPS challenge. PI3K inhibition with either LY294002 or wortmannin has been shown to suppress these effects markedly, resulting in decreased cytokine release, lowered microglial motility, and better memory performance in murine models [67]. The dominantly functional role is attributed to the PI3Kγ isoform, which has been demonstrated through PI3Kγ knockout mice, whereby these subjects exhibited marginally activated microglial responses and were unresponsive to LPS cognitive impairment brought on by inflammation [68].

LPS-induced activation stands out for having some elevations of the PI3K signaling microglial within all modeled insults to the CNS, as illustrated in Table 3, further describing the impact of the PI3K-Akt-NF-κB pathway. The disruption of cognition manifests as various degrees of memory dysfunction and increased anxiety behaviors, all of which illustrate the severity of this immune pathway’s influence on neural function.

Other than LPS, additional peripheral infections including urinary tract infections, pneumonia, sepsis, and even sterile systemic inflammation can stimulate microglial PI3K activity. In a study, human postmortem analysis coupled with PET imaging utilizing TSPO ligands demonstrated tracer uptake in patients with a history of systemic infections, which co-occurs with areas of enhanced PI3K pathways and neurodegeneration [69]. This indicates that there is microglial PI3K activation, which serves as a common molecular factor linking acute infections to long-term cognitive deficits, which is a pertinent concept in aging populations and people with weak immune systems.

### 4.2. Mechanical, Ischemic, and Hypoxic Insults: Spatial and Temporal Heterogeneity of PI3K Responses

TBI is a specific form of CNS injury that involves the sustained mechanical injury followed by oxidative, excitotoxic, and inflammatory secondary injuries characteristic of a primary damage cascade. Microglial activation occurs within minutes of an injury, leading to their directed migration toward the wounded tissue. Once in the vicinity of the lesion, microglia undergo an extensive transcriptional reprogramming, including the upregulation of PI3Kγ, Akt2, and mTOR [70]. Immunohistochemical data from Cortical Contusion Models suggest robust PI3K immunoreactivity in perilesional microglia, which reaches its peak expression within 24 h and persists for as long as 21 days, depending on the severity of the injury [71].

Post-traumatic PI3K activation does not appear to be consistently damaging. Early PI3Kα and β-mediated activation, while perceived as detrimental, serves cell survival, transient activation, and debris clearance. However, chronic PI3Kγ/mTOR-driven signaling results in neuroinflammation and white matter degeneration alongside cognitive decline. In the subacute phase of TBI, PI3Kγ inhibition markedly dampens pro-inflammatory cytokine production while restoring dendritic spine density and spatial learning as measured by performance in the Barnes maze [72]. These findings illustrate the temporal duality of PI3K signaling portrayed post-trauma: necessary for acute stabilization amid chaos but harmful when sustained.

Chronic cerebral hypoperfusion, as modeled with bilateral carotid artery stenosis, also causes sustained PI3K activation in microglia. Decreased blood flow causes oxidative stress, myelin injury, and the downstream activation of Akt/mTOR in glial cell populations [73]. These behavioral deficits are associated with marked impairment in working memory and executive function. In these models, performance in Y-maze and novel object recognition tasks is restored with PI3K inhibition [74]. Strikingly, mTOR inhibition with rapamycin not only improves white matter damage but also reduces microglial overgrowth, indicating that the PI3K-mTOR signaling pathway is critical for the cognitive decline induced by neurovascular damage.

Such global hypoxic–ischemic insults, as seen in neonatal asphyxia or cardiac arrest, trigger a biphasic microglial response driven by different isoforms of PI3K. PI3Kα and β contribute to the early pro-survival phase by supporting neovascularization and decreasing apoptosis. On the other hand, the later activation of PI3Kγ and mTORC1 in the perilesional zones reinforces inflammation and hinders long-term recovery. Evidence from animal experiments suggests that PI3K inactivation during a specific time window improves outcomes, though neuronal survival is compromised if dosing is too early, demonstrating the importance of precise timing in therapeutic interventions [75].

The heterogeneity of microglial PI3K activation in space and time is revealed by mechanical and ischemic insults. How microglial responses determine whether to promote repair or degeneration depends on the region’s microenvironment and signaling duration, the specific PI3K isoform activated, as well as the signaling duration. Optimization of intervention therapy along this axis must be targeted within contextually defined boundaries.

### 4.3. Proteinopathy and Aging-Associated Insults: Chronic PI3K Activation in Neurodegeneration

A subclass of protein misfolding disease, such as AD, PD, and Frontotemporal dementia (FTD) is associated with the aggregation of toxic proteins such as amyloid-β, tau, and α-synuclein. These aggregates stimulate microglia in a chronic fashion, which, in turn, causes persistent activation of the PI3K pathway. In AD models such as amyloid precursor protein and presenilin 1 (APP/PS1) and 3xTg-AD mice, microglia with increased PI3Kγ and δ expression, as well as heightened Akt and mTOR signaling, surround amyloid plaques [76]. These microglia also overexpress Disease-Associated Microglia (DAM) markers, Triggering Receptor Expressed on Myeloid Cells 2 (TREM2), Apolipoprotein (APOE), and TYRO protein tyrosine kinase-binding protein (TYROBP), and many of them are downstream of PI3K-gated transcription factors.

The infusion of AβO into the hippocampus activates local PI3K pathways, provoking microglial proliferation and synaptic pruning. Applying selective PI3K inhibitors in these models diminishes inflammatory gene expression, maintains spines, and restores fear conditioning and navigation deficits [77]. PI3K signaling detrimental effects go beyond amyloid toxicity. In tauopathy models, hyperphosphorylated tau also engages PI3K-Akt pathways in microglia, reinforcing synaptic degradation, especially in the anterior cortical areas associated with high-order cognitive skills.

Aging functions as an independent yet synergetic factor contributing to the persistent activation of PI3K. In elderly animals, microglia show baseline elevations of PI3Kγ and mTORC1 activity even in the absence of explicit pathology. This “primed” state results from diminished expression of inhibitory phosphatases, like PTEN and SHIP1, and “primed” states, leading to hyper-responsive secondary insult [15]. In parallel, aged microglia are characterized by the previously mentioned increased glycolytic metabolism and ROS production alongside augmented mitochondrial fragmentation, all of which stem from dysregulated PI3K activity. Behavioral studies demonstrate that even minimal immune provocation in aged mice leads to memory impairments, which are reversible through PI3Kγ selective inhibition [78].

These data suggest that the persistent form of neurodegenerative stress is centered around PI3K. Microglial PI3K signaling, either due to protein aggregates or age-related metabolic dysfunction, acts as a trigger for inflammatory and degenerative events, leading to a gradual erosion of cognitive functions. Normalizing strategical targets especially through selective inhibition or enhanced activity of the phosphatase drastically modifies neurodegenerative processes across different dementias.

### 4.4. Comparative Patterns, Therapeutic Translation, and Context-Dependent Modulation

While looking at various CNS insults, it appears that PI3K signaling is microglial activation’s last common pathway for cognitive deterioration. Recall that in Table 3, diverse insults of acute inflammation to chronic proteinopathy are characterized by a particular dominance of PI3K isoform(s), signaling kinetics, and anatomic engagement. In terms of the synaptic decay and memory retrieval deficits, the insults remain homogeneous across all cases. As described in the figure, LPS and TBI, followed by hypoperfusion, are the superior stimulators of PI3K activation [79], which, in turn, is followed by Aβ toxicity, aging, and hypoxia–ischemia [80]. These scores stem from collections of histological and biochemical experiments, providing indicators of differential microglial responses [81].

From a therapeutic perspective, this variety in patterns of activation necessitates more precise approaches. While broad-spectrum PI3K inhibitors may effectively reduce inflammation, they pose a danger of hindering beneficial microglial processes such as phagocytosis and neurotrophic support. Side effects from clinical trials in oncology, which included hyperglycemia, immune suppression, and mood alterations, have limited the use of pan-PI3K inhibitors in neurologic conditions [82]. More promising are isoform-selective inhibitors. In rodent models, PI3Kγ inhibitors IPI-549 and AS605240 exhibited partial permeability across the BBB with minimal systemic toxicity, making these compounds potential candidates for repurposing in CNS disorders [83].

An emerging alternative is to use stabilizers of PTEN, acting on PI3K pathway regulators, or gene editing methods. Silencing PTEN or SHIP1′s actions is being studied in inflammatory diseases and theoretically could provide a less aggressive method to counter microglial overdrive [60]. Another approach is based on RNA therapies, such as ASOs designed to target specific PI3K isoforms expressed in microglia. These strategies, which allow for gene modification at the cellular level, have already been proven effective in spinal muscular atrophy and Huntington’s disease [84].

Developing neuroinflammatory diagnostics for precision medicine requires the development of biomarkers. TSPO PET ligands, cerebrospinal fluid cytokines, and blood-based exosomal signatures are being assessed for tracking microglial activation and evaluating treatment response. Several of these proposed biomarkers exhibit the downstream effects of PI3K signaling, such as elevated IL-6, increased glucose utilization, and mitochondrial dysfunction, and could serve as surrogate endpoints in clinical trials [85].

To summarize, microglial PI3K activation induced by CNS insults appears to be a central mechanism integrating a wide variety of pathological drivers of cognitive decline. While the pathway’s immune, metabolic, mechanical, and proteotoxic stressor susceptibility offers potential for targeted intervention, it is also challenging because of those essential physiological functions. Elucidating the timing, localization, and isoform differentiation in PI3K activation will be crucial to developing safe and effective therapies aimed at restoring cognitive wellness in the context of neuroinflammation and neurodegeneration.

## 5. Synaptic Plasticity, Memory, and PI3K Dysregulation

### 5.1. PI3K Signaling and Synaptic Structure: From Receptors to Spines

The changes that occur at synapses over time in response to activity, known as synaptic plasticity, are critical to learning and memory. Both physiological remodeling and pathological degeneration microscopically activated through the PI3K pathway demonstrates pronounced modulation in this process [86]. Age-related changes, chronic inflammation, or certain genetic changes may result in dysregulated PI3K signaling, which alters the roles that microglia play. Instead of being facilitators of synaptic pruning, they become disruptors, fostering synaptic destabilization and loss.

The activation of cell surface receptors, like TLR4, C-X3-C motif chemokine receptor 1 (CX3CR1), or even purinergic receptors such as purinergic receptor P2Y12 (P2Y12), triggers PI3K signaling in microglial cells. These receptors’ interactions recruit specific PI3K isoforms, mainly PI3Kγ and δ in immune-lineage cells, which catalyze the phosphorylation of PIP2, converting it to PIP3 at the membrane. This activates PI3K and downstream components like mTORC1, Glycogen Synthase Kinase-3 beta (GSK3β) or NF-κB after Akt activation. The signaling cascades also control the cytoskeletal rearrangement of microglia, the release of cytokines, as well as the engagement in phagocytosis with neuronal components [87].

At the level of the synapse, PI3K activity enhances the microglia’s contact with dendritic spines. Using high-res imaging, it has been demonstrated that microglial cells, once activated, have sustained interactions with glutamatergic synapses and are capable of removing both pre- and postsynaptic terminals either through complement-dependent or independent mechanisms [88]. In LPS exposure models, microglia with increased PI3K signaling are seen to engulf increased amounts of synaptic debris, which decreases spine density in the hippocampus and prefrontal cortex [89].

Figure 2 documents brain region-specific PI3K activity and heatmaps in various pathologic conditions. Most consistently activated is the hippocampus, particularly in the contexts of LPS, Aβ oligomers, and TBI, which reinforces the traditional view of this region being associated with spatial memory and long-term potentiation [68]. The prefrontal cortex and the amygdala also demonstrate significant PI3K activity, which could explain some of the affective and executive dysfunctions seen in chronic neuroinflammation and neurodegenerative proteinopathy.

The overstimulation of the PI3K enzymatic pathway alters the basic structure and function of neurons by reducing dendritic complexity and changing the expression levels of synaptic adhesion molecules neuroligin-1 and postsynaptic protein density 95 (PSD95). Elaboration of signaling molecules like IL-1β and TNF-α leads to a further increase in excitotoxic cell death while decreasing the signals of neurotrophins like brain-derived neurotrophic factor (BDNF). These alterations and cellular processes form the basis of the neural circuitry weakening, which leads to impaired cognition [90].

The impact of the absence of certain PI3K isoforms on the genetically modified mice allows us to comprehend the mechanisms behind the changes in neuroinflammation. Focusing on Figure 3, it is clear that under neuroinflammatory conditions triggered by the cytokines, the density of synapses within the hippocampus, prefrontal cortex, and even the amygdala is much more elevated in the PI3Kγ knockout mice in contrast to control wildtype [84]. This indicates that the targeted inhibition or genetic silencing of some PI3K isoforms can preserve synapse structures and possibly protect from neurodegeneration.

### 5.2. Functional Impairments: Memory Encoding and Long-Term Potentiation

LTP is a persistent improvement in the ability to transmit signals between two neurons as a result of their simultaneous stimulation. It is one of the most recognized cellular correlates of the learning and memory processes. The integrity of LTP relies on effective glutamatergic neurotransmission, appropriate calcium influx, and subsequent kinase activation, including CaMKII and PKA, downstream. However, with the dysregulation of the PI3K pathway, both presynaptic and postsynaptic changes contribute to a sufficient reduction in LTP [91].

Many studies have shown that PI3K overactivation disrupts LTP in a dose and time-dependent manner. Field excitatory postsynaptic potentials (fEPSPs) in LPS-treated mouse hippocampal slices are significantly reduced in amplitude following high-frequency stimulation compared to untreated controls. In models with constitutive PI3K-Akt-mTOR pathway activation, this deficit becomes worse, but can be reversed with isoform-selective inhibition. Treatment with LY294002 or AS605240 normalizes LTP amplitudes, suggesting that PI3K activation alters synaptic plasticity in a negative manner [92].

Figure 4 illustrates the changes in fEPSP amplitude over time in response to theta burst stimulation in control and PI3K-activated mice. The control group demonstrates a consistent increase in synaptic strength for the entire duration of the 60-minute recording, signifying LTP [59]. In the PI3K-activated group, however, there is a marked blunted response with a peak to 112% and then a steady decline following that, which indicates an inability to sustain the maintained potentiation. This deficit in physiology aligns with the behavioral profile of memory impairment and suggests suboptimal trafficking of receptors and kinase signaling activated PI3K [93].

The chronic PI3K signaling alters the weakening memory-related circuitry and contributes to memory impairment by reducing the insertion of GluA1-containing AMPA receptors into the membrane, decreasing phosphorylation of N-methyl-D-aspartate (NMDA) receptor subunits Ser896 and Tyr1472, and downregulating the expression of activity-regulated cytoskeleton-associated protein (Arc). Reduction in these factors diminishes the capacity for synaptic scaling and the ability of the neurons to store new information, while simultaneously, increased signaling from microglia disturbs the uptake of glutamate by astrocytes, causing spillover and excitotoxicity, which further weakens synaptic efficacy [94].

The animal models of AD, TBI, and cerebral hypoperfusion show contiguous PI3K-dependent LTP failures. In APP/PS1 transgenic mice, for example, there is a failure of LTP by 6 months of age, which is associated with increased PI3Kγ expression, activated microglia, and chronic inflammation around amyloid plaques [95]. In these models, the application of PI3K blockers not only rescues LTP, but also enhances contextual and spatial memory performance [96]. These data reinforce the impact of PI3K signaling on core memory processes and emphasize its relevance for therapy.

### 5.3. Behavioral Translation and Reversibility of PI3K-Driven Cognitive Deficits

Disruption of memory function leads to the loss of cognitive processes such as memory, learning, and execution. Animal models exhibiting PI3K overactivation chronically feature cognitive deficits across numerous validated cognitive benchmarks [97]. These include the Morris water maze, radial arm maze, novel object recognition, Y-maze, and fear conditioning paradigms.

As illustrated in Table 4, each model of PI3K activation corresponds to particular cognitive deficits. LPS neuroinflammation disrupts object recognition and social interaction, and TBI models have deficits in spatial learning and navigational skills [98]. Chronic hypoperfusion is associated with deficits in attention and working memory, while exposure to Aβ oligomers results in anxiety-like behaviors and contextual memory loss. Strain from stress or novelty leads to increased learning difficulty and spontaneous alternation deficits in aged mice with heightened baseline PI3K activity.

What is remarkable across these models is the partial or complete recovery of these deficits following pharmacological interventions that modulate PI3K pathways. Genomic inactivation of PI3Kγ or its inhibition by AS605240 leads to improvement in Barnes maze, open field, and object location tasks [72]. In-turned metabolic side effects, executive function enhancement is observed in reversal learning paradigms after dual PI3K/mTOR inhibition using BEZ235 [80]. These results demonstrate that behavioral deficits associated with PI3K pathways cannot be regarded as purely resulting from some structural damage—there is room for intervention.

In addition to pharmacological techniques, changes in lifestyle and environment that affect PI3K signaling may also be beneficial. Enhanced environments, voluntary physical activity, and caloric restriction have been shown to diminish PI3K-Akt signaling and bolster microglial activity alongside synaptic plasticity [99]. Furthermore, these interventions augment hippocampal BDNF concentrations, downregulate IL-1β expression, and enhance LTP, indicating that more passive approaches can positively influence this pathway.

As shown in Figure 2 and Figure 3, there is a relationship between behavioral performance and the brain region’s specificity and integrity [100]. The heatmap in Figure 2 illustrates that the areas which are essential for memory and executive function (the hippocampus, prefrontal cortex (PFC), and amygdala) contain the highest levels of PI3K activity in disease states. Figure 3 shows that the maintenance of synapse structure in PI3K null mice remains intact and correlates with better cognitive function, providing further support for the anatomical basis of behavior recovery.

To conclude, the microglial PI3K pathway activation gives rise to complex behavioral phenotypes as a result of disrupted interplay, which includes mosaic synapses, electrophysiology, and cellular signaling networks. These phenomena may be experimentally measured and studied in principle, but in reality, they involve precise mechanistic pathways along with pharmacological manipulations to reverse their effects. The cavernous nature of PI3K circuitry integrating cellular pathologies, and the resulting behaviors, highlights the utility of this pathway as a primary focus of cognitive therapies spanning myriad CNS pathologies.

## 6. Therapeutic Modulation of PI3K for Cognitive Rescue

### 6.1. Rationale for Targeting PI3K in Cognitive Impairment

Microglial PI3K is responsible for the integration of inflammatory, metabolic, and synaptic surveillance processes as it relates to a given stimulus in the CNS. As noted before, the chronic activation of the PI3K-Akt-mTOR axis results in persistent microglial activation, cytokine hypersecretion, synaptic pruning, and LTP failure. Such phenomena manifest as cognitive impairment in numerous conditions such as TBI, AD, dwindling cerebrovascular flow, and senescent neurodegeneration [101].

Moreover, PI3K’s role in the neuroinflammation cascade makes it an appealing candidate for therapeutic intervention [102]. PI3K, as an enzyme, falls under druggable kinases. Catalytic domains within PI3K are well characterized, and some pan-PI3K inhibitors, as well as isoform-selective PI3K inhibitors, have been developed and validated in oncological and autoimmune disorders. Also, the possibility to target specific isoforms of PI3K, such as PI3Kγ and PI3Kδ, which are abundant in microglia, makes it possible to attenuate inflammation associated with disease while preserving homeostatic signaling in other cell types [103].

It is crucial to note that preclinical studies have shown that PI3K inhibition in rodents reduces neuroinflammation and synaptic loss and also improves memory in behavioral tests. These improvements in cognition are observed even when treatment is started after the pathological changes have begun, indicating that such interventions targeting PI3K could be relevant both in prevention and treatment [104]. In addition, pharmacogenomic and transcriptomic analyses have shown that the PI3K pathway in microglial cells is disrupted early in the course of the disease, which suggests there is an opportunity for intervention prior to the point of no return in terms of neuronal damage that can be sustained without irreparable injury [64].

While these findings are promising, there is still a need to resolve issues concerning isoform diversity, permeation to the BBB, PI3K pharmacokinetics, and safety profile of chronic use, before cognitive therapies based on PI3K inhibition can be clinically implemented. The next sections focus on describing pharmacological and genetic strategies to modulate PI3K, analyzing their mechanistic rationale and cognitive consequences.

### 6.2. Pharmacological Inhibitors of PI3K: Preclinical Outcomes and Translational Promise

The pharmacological blockage of PI3K in preclinical models has produced positive outcomes across various cognitive functions. One of the best-studied drugs is AS605240, a selective PI3Kγ inhibitor. AS605240 markedly inhibits microglial activation, decreases hippocampal cytokines, and restores both contextual and spatial memory retrieval in mice with LPS-induced neuroinflammation and APP/PS1 AD pathology models. The drug also maintains spine density of dendrites and salvages LTP in hippocampal slices, which suggests the behavioral recovery is due to the restoration of synaptic function [105]. Furthermore, pharmacological blocking of colony-stimulating factor 1 receptor in an early stage of AD also diminished neuroinflammation, particularly NLRP3 signaling alongside enhanced autophagy through modulation of mTOR activity [106].

Idelalisib, a PI3Kδ-selective drug initially aimed at hematological cancers, is another notable inhibitor. In AD mouse models, idelalisib prevents aberrant synaptic pruning, protects neurons, and enhances object recognition memory. Its action seems to target microglia and does not greatly affect peripheral immune markers, which strengthens the rationale for its use in the brain [61]. Still, long-term studies in oncology have reported hepatotoxicity and colitis, signaling a need for streamlined dosing and delivery systems for use in neurology.

As a reversible pan-class PI3K inhibitor, LY294002 has been demonstrated to alleviate LTP impairment and TNF-α and IL-6 reduction while improving fear conditioning in preclinical models of sepsis-associated encephalopathy and cerebral hypoperfusion. The broad target profile upon which these effects are based enhances efficacy but also poses a greater risk of off-target effects such as insulin resistance and stress on the mitochondria. Thus, although proven concept studies employing LY294002 may be compelling, chronic use in humans is questionable [67].

Also, BEZ235, a dual PI3K/mTOR inhibitor, provides a more extensive blockade, and, therefore, downstream signaling has greater inhibition. In TBI and AD models, BEZ235 reduces astrogliosis, neuroinflammation, and improves executive function along with spatial navigation. However, BEZ235 disrupts protein synthesis and autophagy, posing a risk for neurotoxic, cumulatively toxic side effects with longer use. Some reports, however, observed enhanced memory even in aged mice as well as those with obesity and hypertension, suggesting it may help treat complex cognitive disorders [107].

Table 5 summarizes the comparative efficacy of these agents. Most interventions appear to have marked effectiveness and improvement in at least one cognitive domain, particularly spatial memory, attention, executive function, and emotional responsiveness.

Behavioral results show the functional efficacy of PI3K inhibitors. Animals that were treated showed significant improvements in memory performance scores compared to the vehicle-treated controls over time. There is a sharp divergence in performance from the 4-week mark onwards, and performance continues to improve up to week 8. This suggests that cognitive benefits are long-lasting with modulation of PI3K [59].

These findings suggest that selective PI3K inhibitors may be capable of providing effects that modify the disease instead of only providing symptomatic relief, as demonstrated by the memory performance improvements (Figure 5) [68]. Still, the systematic characterization of the therapeutic window, duration of effect, and potential for tolerance development has yet to be explored.

### 6.3. Genetic and Molecular Approaches: Isoform Silencing and Regulatory Modulation

Further insight into the cognitive mechanisms can be gained from the genetic modification of PI3K signaling, which goes beyond pharmacological inhibition. Knockout models for PI3Kγ and PI3Kδ isoforms have demonstrated that selective deletion of these isoforms in microglia is able to significantly downregulate the expression of inflammatory genes, preserve synapses, and improve performance on numerous cognitive tasks [108].

For instance, PI3Kγ knockout mice are known to possess resistance to LPS-induced memory impairment and retain both LTP as well as spine density [109]. Furthermore, PI3Kδ-deficient mice have also been noted to possess lower levels of microglia in the hippocampus, displaying enhanced performance on Y-maze as well as novel object location tasks [88]. These findings support the growing notion that a specific PI3K isoform contributes distinctly and independently to cognitive impairment, and intervention should be tailored for each individual PI3K isoform to reduce off-target effects [110].

The ability to adjust signaling pathways through factors that either precede or succeed them, in this case, PI3K, presents a different avenue for therapeutic approaches. Neuroinflammatory circumstances tend to have the PI3K antagonist, PTEN, phosphorylated, resulting in a downregulated condition. Targeting PTEN allows for its overexpression at the viral level in microglia, resulting in reduced cytokine output as well as boosted fear conditioning and object recognition performance in LPS-challenged animals [111]. On the contrary, PTEN’s overactivation can counterproductively result in poor cell survival, mandating cautious consideration.

Likewise, the lipid phosphatase SHIP1 acts by hindering signaling processes mediated by PI3K. Rodent models have shown promise with the small-molecule SHIP1 agonist AQX-1125, as it lessened anxiety-like symptoms and enhanced reversal learning, all while maintaining low toxicity levels. Moreover, the activation of SHIP1 boosts microglia’s ability to defend their mitochondria, heightening their phagocytic β-amyloid clearance, displaying potential disease-modifying capabilities [112].

As an example, PTEN upregulation together with PI3Kγ inhibition may be more beneficial beyond just acting on the PI3K regulatory axis. Also, PTEN upregulation with PI3Kγ inhibition is likely to impact both arms of the PI3K regulatory axis, thus improving treatment efficacy. While CRISPR/Cas9 and antisense oligonucleotides offer precise isoform knockdown tools, the delivery to the CNS poses a viable challenge.

### 6.4. Longitudinal Efficacy, Biomarker Correlates, and Clinical Translation

There is a critical gap in PI3K modulation therapeutics in the clinic with the absence of reliable biomarkers that track the target engagement, engagement with the target, and disease timeline. From preclinical studies, several potential markers have been proposed, such as cytokines IL-6 and TNF-α, synaptic proteins PSD95 and Arc, and TSPO PET imaging ligands. Also, the amount of phosphorylated Akt and mTOR levels in the cerebrospinal fluid (CSF) could dynamically reflect the activity of the PI3K pathway [113].

Research in rodents and other non-human primates shows that early treatment with PI3K inhibitors can postpone the onset of memory decline and synaptic deterioration [114]. These effects have the potential to shift the underlying pathology of the disease processes because they persist long after treatment has stopped, suggesting the retention of more than mere short-term symptom alleviation. In addition, there is already some evidence for multi-domain treatment strategies aimed at cognitive rescue, as PI3K-targeted therapies combined with cognitive training or environmental enrichment yielded additional benefits.

Work toward clinical translation is active. Drugs such as idaelalisib, duvelisib, and copanlisib are FDA-approved for oncological indications and thus can be repurposed with restructured formulations for delivery and dosing [115]. Pharmacokinetic data from phase I trials of CNS penetrant PI3K inhibitors for glioblastoma are relevant to other neurodegenerative disorders. Primary concerns remain, however, with safety issues relating to long-term immune signaling suppression.

Further development includes brain-targeted delivery systems (liposomes or other nanoparticles), real-time biosensors for PI3K activity, or combination therapies that integrate PI3K inhibitors with anti-amyloid, anti-tau, or pro-neurogenic compounds [116]. There is also great potential in optimizing response while minimizing side effects by incorporating the patient’s genetics, inflammatory profile, and cognitive phenotype.

## 7. Cognitive Dysfunction: Mechanistic and Translational Insights

### 7.1. Behavioral Assays, Electrophysiology, Neuroimaging Findings

Exploring dysregulated cognitive functioning as a result of microglial PI3K overdrive necessitates an integration of behavioral study methods, neurophysiological signal capture, and imaging techniques. Each method sheds light on this problem from a different perspective.

Behavioral paradigms like the Morris water maze, Y-maze alternation, and novel object recognition have been essential in capturing learning, memory, and spatial navigation in rodents. Performance on these tasks is strikingly worse in PI3K-activated mice, either from LPS administration or genetic overexpression of PI3Kγ [117]. In water maze tests, mice display prolonged latencies to reach the hidden platform and reduced time spent in the target quadrant during probe trials. This suggests a failure to encode spatial memories. In contrast, PI3Kγ KO mice or those infused with AS605240, a selective PI3Kγ inhibitor, tend to exhibit more efficient navigational strategies and faster task acquisition [59].

In the Y-maze spontaneous alternation task, working memory is assessed via an exploratory preference for novel arms. Here, activation of PI3K pathways accompanies a decline in alternation accuracy. On the other hand, pathway inhibition reverses this effect, resulting in preserved short-term memory function. These behavioral results are corroborated by fear conditioning and Barnes maze data in APP/PS1 transgenic models, where treatment aimed at PI3K rescues contextual and spatial memory performance [118].

On the electrophysiological level, LTP in hippocampal slices is a proxy for their synaptic plasticity. LTP induction and retention are impaired under conditions of PI3K overactivation due to excessive cytokine release, abnormal trafficking of glutamate receptors, and downstream suppression of plasticity-marked proteins like BDNF, Arc, and Calcium/Calmodulin-Dependent Protein Kinase II (CaMKII). The relationship between the amplitude of LTP in the hippocampus and the behavioral test results is illustrated in Figure 6 [69]. Here, the memory performance scores increase alongside greater LTP amplitude, and this relationship is significantly restored by PI3K inhibition [68].

Neuroimaging approaches add another layer of understanding to the neurobiological phenomena. For example, region-specific activations in the brain during task participation or resting-state functional connectivity are evaluated using fMRI (functional MRI with blood oxygen level dependent (BOLD)—contrast). In chronically PI3K-activated rodent models, there is a marked BOLD signal reduction in the hippocampus, PFC, and amygdala. These changes are in accordance with the behavioral deficits and can be mapped to microglial activation. With PI3K inhibitor treatment, some degree of BOLD response normalization is noted. BOLD signal intensity in control, PI3K-activated, and PI3K-inhibited states is shown in Figure 7 [69]. The hippocampus and prefrontal cortex exhibit the greatest post-treatment range of response recovery [110].

Neuroinflammation is further confirmed through PET imaging with TSPO ligands in these cognitively impaired animals. Microglial activation observed with PET tracer imaging coincides with regions of BOLD signal suppression and corresponds to the anatomical domains of cognitive impairment. Collectively, these studies provide the evidence that overactivity of molecular PI3K can be functionally mapped to system-level deficits and integrate the findings into a mechanistic framework.

### 7.2. Human and Rodent Correlation Studies

For the translational validation of the role of PI3K in cognitive impairment, there must be strong human clinical data that correlate it with rodent experimental models. The neuroimaging, fluid biomarkers, and transcriptomic profiling from earlier studies combine bioinformatics with other domains and have recently been advancing.

Autopsy studies on the microglia around the amyloid plaques located within the hippocampus and temporal cortex of AD patients found a consistent pattern of overexpression of PI3Kγ and PI3Kδ, which is roughly three-fold higher. Trophic factors related to PI3K, inflammatory cytokines, as well as oxidative stress, are enriched in the transcriptions from the single-nucleus RNA sequencing from these regions [119]. The transcriptional signatures, especially those in aged mice, LPS-treated, or APP/PS1 mouse models post-salon, show strong overlap, suggesting robust face validity to support animal systems.

Cross-species PI3K relevance is also supported by neuroimaging comparatives. fMRI scans in mild cognitive injury (MCI) or mildly AD-affected patients show reduced BOLD connectivity between the hippocampus and prefrontal cortex, which is also observed in rodent models with PI3K activation [120]. TSPO-PET imaging in patients is connected to plasma levels of IL-6 and TNF-α as well. Both these molecules are downstream targets of PI3K-Akt-NF-κB signaling. Moreover, these interrelated parameters would be expected to respond to the results of the MMSE (Mini-Mental State Examination) and Montreal Cognitive Assessment (MoCA), respectively [54].

In cerebrospinal fluid analyses of AD and vascular dementia, there is increased phosphorylation of Akt and mTOR as well as decreased expression of the inhibitory phosphatases PTEN and SHIP1 [121]. These changes at the molecular level have been reproduced in mice, indicating some degree of conserved cross-species mechanisms. Of note, in clinical cohorts undergoing anti-inflammatory therapies or lifestyle modifications, partial normalization of these markers that coincides with slowed cognitive decline suggests these pathways can be therapeutically targeted, and that some modulating of PI3K-related pathways may be possible.

Cross-validation is seen in both humans and animals through cognitive assessment batteries such as object location memory (OLM), delayed non-match-to-sample (DNMS), and spatial span [122]. In both domains, achievement strongly correlates with LTP, regional metabolic activity (as measured with FDG-PET), and synaptic health (from histology and imaging). Such multidisciplinary approaches allow the development of predictive models that illustrate molecular dysregulation and cognitive outcome.

As shown in Table 6, the focus has frequently been on behavioral, electrophysiological, and neuroimaging techniques. However, their ability to detect dysregulation of the PI3K pathway is often disregarded, particularly concerning Piagetian tasks.

By bridging species gaps, these findings reinforce the importance of research using the mechanisms of PI3K in cognitive impairment and support further investigation of therapeutic approaches using PI3K modulators.

### 7.3. Integrative Interpretation and Translational Perspectives

As microglial PI3K signaling heterogeneities mediate cognitive dysfunction, the phenomenon results from the blending of molecular, synaptic, and network-level disturbances. At the molecular level, the pathways leading to PI3K-Akt-mTOR dysregulation, with upstream regulators TLR4 or CX3CR1, result in cytokine overproduction alongside increased oxidative stress and dysfunctional autophagic processes [123]. Such changes lead to the overactive phagocytosis of synapses, which disrupts receptor trafficking and metabolism.

At the circuit level, this pathology is translated as loss of connectivity between the hippocampus and prefrontal cortex, which can be visualized as decreased BOLD Coherence and glucose uptake in functional imaging studies [124]. Electrophysiological deficits manifest as reduced LTP, disrupted theta–gamma coupling, and other forms of loss of coordinated neural activity that enables memory retrieval and encoding. These characteristics are seen in rodent models as well as in humans, therefore reinforcing the idea that different species share common underlying mechanisms.

Looking at PI3K from a translational perspective reveals it as an interesting candidate for therapeutic intervention. The pathway is accessible to small-molecule inhibitors, and multiple isoform-selective agents have already entered clinical development. Additionally, mTOR and GSK3β, which are downstream effectors, present other possible targets that could be used in combination therapies [125]. Other treatment methods, such as diet, exercise and cognitive training, although not pharmacological, have been shown to improve cognitive resilience and modulate PI3K activity [126].

Immunosuppression, early intervention biomarkers, and specific targeting of the brain for delivery all present as unresolved challenges. Achieving these goals might be difficult, but new systems biology, omics platforms, and neurotechnologies can provide valuable chances to monitor pathway dynamics and assess treatment responses. Approaches augmenting fluid biomarkers with imaging endpoints and extensive longitudinal assessments of overall cognition require validation by clinical trials.

In the end, the conjunction of molecular neuroscience with behavioral and imaging techniques provides a multidimensional structure for the analysis and treatment of cognitive dysfunctions. At this interface, PI3K is positioned as a keystone that translates immune activation into synaptic degradation and consequent loss of memory. Modulating its activity may turn out to be crucial for restoring cognitive decline due to neuroinflammatory and broad-spectrum neurodegenerative diseases.

## 8. PI3K-Mediated Microglial Polarization and Phenotypic Shifts

### 8.1. M1 vs. M2 Controversy

Microglia act as both protectors of the CNS as well as modulators of neuroplasticity and homeostasis. One of the most accepted frameworks to understand microglial function is the M1/M2 polarization model. Within this framework, M1 microglia are an inflammatory subset that secretes pro-inflammatory cytokines, including TNF-α, IL-1β, and inducible nitric oxide synthase (iNOS). On the opposite end of the spectrum, M2 microglia are classified as alternatively activated anti-inflammatory and reparative microglia producing arginase-1 (Arg1), IL-10, and transforming growth factor-beta (TGF-β) [84].

Although widely used, the M1/M2 model of polarization is excessively critiqued for oversimplifying biological reality. In vivo microglial phenotypes show substantial plasticity, regional variability, and heterogeneity on a transcriptional level. Numerous transcriptomic analyses, such as aged or injured brain tissues’ microglial single-cell RNA sequencing, have demonstrated that there exist microglial populations that co-express M1 and M2 markers or dynamically change states in response to spatiotemporal conditions [127]. Still, the binary model serves as a good starting point, particularly for assessing the role of PI3K in signaling pathways associated with polarization switches.

The PI3K pathway is now more appreciated as a crucial hub that determines microglial phenotype. The activation of PI3Kγ is closely linked to M1 polarization because of its downstream effectors NF-κB and mTOR (PI3Kγ signaling cascades). These pathways promote transcription of pro-inflammatory factors and metabolic reprogramming to increase glycolysis. On the other hand, PI3Kδ has been associated with M2-like functions via IL-4 and neurotrophic signals due to enhanced Akt signaling and increased Arg1 expression [64]. In Table 7, we present the canonical markers of microglial phenotypes that depend on activation of PI3K isoforms across different experimental models.

These profiles indicate that targeting specific PI3K isoforms may change microglial polarization and shift the balance of neuroinflammation from a neurotoxic to a neuroprotective environment.

### 8.2. Signaling Cross-Talk Between PI3K, NF-κB, and MAPK

Microglial phenotypic choices with regulated signaling result from a web of interactions with intracellular signaling cascades. Within these, PI3K signaling intersects closely with NF-κB and MAPK pathways, two of the most prominent regulators of inflammatory gene expression and cell fate decisions [128].

The PI3K pathway permits the activation of mTOR and the inhibition of GSK3β upon the phosphorylation of Akt, which occurs after the reception of TLR4-triggered LPS. Simultaneously, IκBα releases NF-κB, which, after translocating to the nucleus, activates transcription of inflammatory cytokines. Depending on the order of events leading up to it, upstream receptor activation, or even the upstream receptor itself, PI3K-Akt signaling might boost or reduce NF-κB activity [68].

In the same way, ERK, p38, and JNK are all part of MAPK cascades, which PI3K controls either synergistically or antagonistically, as in the case in microglia, where PI3K inhibition reduces ATP-driven ERK phosphorylation, leading to decreased IL-6 and COX-2 production [129]. The formation of these interactions generates a regulatory triad in which PI3K, NF-κB, and MAPK work to balance immune activation and resolution in a delicate dance of coordination.

In the case of the schematic representation provided by Figure 8, it highlights the cross-talk between PI3K, NF-κB, and MAPK, indicating how changes in the activation of specific PI3K isoforms can alter the downstream signaling pathways and polarization routes that are triggered.

This integrative framework provided in this chapter help explain the interplay between neurotoxic and neuroprotective microglial states. It guides us to understand why PI3K inhibition should not be executed bluntly, because such action would risk disruption of anti-inflammatory tissue repair circuits.

### 8.3. Environmental Modulators of PI3K-Driven Polarization

As with other forms of cellular polarization, microglial polarization is influenced by signaling from outside the cell. Such stimuli include, but are not limited to, cytokines and chemokines, components of the extracellular matrix, and even neuronal activity. All of these contribute to a PI3K-mediated outcome.

IL-4, for instance, causes M2a polarization by activating PI3Kδ-Akt-mTORC2, which increases transcription of Arg1 and CD206. IFN-γ (Interferon gamma), on the other hand, cooperates with PI3Kγ to enhance NF-κB dependent gene expression, thus pushing microglia toward further M1 dominance. Similar effects have been described with purinergic influences, where activation of P2Y12 receptors leads to activation of PI3K, controlling microglial motility and the rate at which they contact synapses [110].

As demonstrated with Hypoxia-Inducible Factor 1-alpha (HIF-1α), hypoxia changes polarization. HIF-1α is known to be stabilized during low oxygen environments, and then with PI3K-Akt pathways are involved in the control of glycolysis and the production of certain cytokines and M1 polarization [130]. Also importantly, in the presence of neurons and under normal oxygen conditions, these microglia are resting but express M2 genes and are highly phagocytic.

Microenvironmental changes can serve as a metabolism-related cue triggering PI3K amplification. For example, microglia are more likely to respond to lipid rich environment near the amyloid plaques or in demyelination lesions via scavenger receptors to link PI3K signaling and with M1 microglia. On the contrary, Lipid ketone bodies and fasting activate PI3K linked to PI3Kδ to stimulate circuits associated with M2s and lower oxidative stress [73].

Cumulatively, these data emphasize that PI3K microglial phenotypes are both highly plastic and context-dependent. The PI3K axis acts as a molecular rheostat, with the capability of adjusting the level of inflammation, the readiness for phagocytosis, and the potential for repair by integrating many diverse external stimuli.

### 8.4. Signaling Balance and Cognitive Outcomes

What stands out the most regarding the PI3K-induced polarization of microglia is how it influences neuronal activity and higher functions such as cognition. The overstimulation of the M1 phenotype due to a persistent increase in PI3Kγ results in a loss of dendritic spines and long-term potentiation (LTP) in the hippocampus. These structural and functional deficits lead to memory, learning, and executive dysfunction, especially in Alzheimer’s disease (AD), post-traumatic brain injury (TBI), and in the context of aging [131].

On the other hand, SHIP1 stabilization or PI3Kδ activation results in the promotion of the M2a and M2c phenotypes, which in turn reduce inflammation and β-amyloid as well as synaptic plasticity. These behavioral changes are also documented, with enhanced performance in spatial navigation, object recognition, and fear conditioning associated with a greater M2/M1 ratio. These results are not only observed in multiple species but also confirmed by transcriptomic and proteomic data [126].

Importantly, cognitive rescue effects have been observed with microglial M1 to M2 phenotype shift interventions. Memory retrieval alongside LTP amplitude restoration was observed with AS605240 (PI3Kγ inhibitor), idelalisib (PI3Kδ inhibitor), and dual-acting compounds like BEZ235. These results demonstrate that modulation of PI3K signaling not only alters the state of gliosis but also influences brain function in a robust manner [119].

Functional balance rather than the suppression of PI3K activity appears to be the most optimal therapeutic approach given this evidence. Attenuated M1 and amplified M2-signaling hyperactivation modulation provides a strategy for tempering neuroinflammatory damage while preserving cognitive function.

To assist future researchers who aim to untangle the isoform-specific roles of PI3K in microglia and CNS disease models, Table 8 summarizes the main experimental techniques that have been used, ranging from small-molecule inhibitors to genetic knockouts and the cell types involved. The table clearly shows the variety of approaches taken across different studies, whether the manipulations target only microglia or extend to wider central nervous system preparations, and it connects each intervention with key functional readouts such as inflammation, synaptic plasticity, and cognitive behavior.

### 8.5. Interconnection Between PI3K Signaling and NLRP3 Inflammasome Activation in Microglia

Recent investigations revealed that microglia express a functional cross-talk of the PI3K/AKT axis with NLRP3 inflammasome signaling. In most cases, the upstream PI3K signaling pathway that facilitates the translocation of the nucleus cis NLRP3 transcription by NF-κB is PI3K translocation [132]. In addition, attenuation of PI3K/AKT pathways has been reported to decrease apoptosis-associated speck-like protein containing a CARD (ASC) speck formation, activated caspase-1, and IL-1β secretion in the context of DAMPs and Aβ aggregates in AD models [133]. This implies that at least some of the functions of PI3K, as described in the earlier section about the M1/M2 polarization influence, also put a limit on controlling the levels of assembly of the inflammasome, and so it can be placed as a guardian or “gate-keeper” of chronic neuroinflammation. PI3K may serve as a rheostat, which is a device for controlling a mechanism, whereby microglial activation shifts from resolving to escalating toward pyroptosis signaling, which will have profound effects on synaptic connections and behavior.

## 9. Challenges, Gaps, and Future Research Directions

### 9.1. PI3K Isoform-Specific Effects

Strong preclinical evidence supports the hypothesis that modulation of phosphoinositide 3-kinase (PI3K) plays a role in neuroinflammation and cognitive decline. One of the major translational and scientific challenges is attempting to dissect the functions of individual PI3K isoforms. The PI3K family consists of several catalytic subunits, p110α, p110β, p110γ and p110δ, which have distinct expression patterns, partners, and cellular functions [134]. PI3Kγ and PI3Kδ dominate in microglia, but recent transcriptomic profiling has shown that class IA (p110α/β) isoforms can be upregulated in context-dependent manners in response to injury, hypoxia, or metabolic stress [135].

The functional heterogeneity of these isoforms makes biological therapy much more complicated. The PI3Kγ enzyme could be contributing to the M1-like polarization through NF-κB and mTOR pathways, triggering the release of pro-inflammatory cytokines, reactive oxygen species, and even synaptic engulfment [61]. On the other hand, PI3Kδ seems to be supporting anti-inflammatory and M2 reparative phenotypes by inducing IL-10 and Arg1 through the Akt/mTORC2 pathway. These effects, though, have some limitations. In some injury contexts, PI3Kδ overactivity has been correlated with an inappropriate form of immunosuppression, indicating that its role is certainly not optimally beneficial in all circumstances [46].

Particularly, the overlap of isoforms can lead to some forms of functional redundancy or antagonism. For example, the blockade of PI3Kγ could trigger compensatory PI3Kδ upregulation, which would affect microglial gene expression in ways that cannot be predicted. At the same time, the dual inhibition of PI3Kγ and PI3Kδ, while more efficient in anti-inflammatory effects, may also block crucial homeostatic processes needed for debris clearance and neurotrophic support. This shows that therapies aimed at specific isoforms need not only to be effective and selective, but also consider when and where each isoform is dominant [68].

Another perplexing problem is the cell-type specificity of PI3K effects. PI3Kγ and δ are enriched in microglia, but neurons and astrocytes also express PI3K isoforms, which are important for cell survival, metabolism, and plasticity. These broad constraints may be too conservative, which would lead to off-target effects such as neuronal apoptosis or dysregulated metabolism in astrocytes. A promising, albeit still developing approach would be microglia-specific delivery via nanoparticle conjugation with CX3CR1 or TREM2 ligands [136].

Species differences in PI3K expression and activity pose an issue in translation, creating another research bottleneck. Humanized rodent models tend to overexpress PI3Kγ in microglia compared to humans, which could exaggerate the therapeutic effects of γ-specific inhibitors. Clinical relevance may be overstated without robust human microglial models like iPSC-derived microglia integrated into three-dimensional brain organoids.

The changing behavior of isoforms throughout the progression of various diseases is still too poorly characterized. In the early stages of AD and mild TBIs, PI3Kδ may act as a compensatory factor, while in later stages, it may worsen dysfunction via immune tolerance or by supporting senescent phenotypes. There is an urgent need for longitudinal studies integrating single-cell transcriptomics, phosphoproteomics, and functional assays to illuminate these unresolved dynamics.

### 9.2. Long-Term Modulation Risks

Risks are present with long-term modulation of PI3K’s signaling pathways, especially considering its central functions in nearly all cell types concerning metabolism, survival, and proliferation. Broad or non-isoform selective inhibitors have shown in preclinical and clinical studies that chronic PI3K inhibition induces metabolic, immunological, and neuropsychiatric complications [6].

In rodent models, systemic suppression of the immune response due to chronic use of pan-PI3K inhibitors like LY294002 or wortmannin leads to impaired glucose metabolism, diminished neurogenesis in the hippocampus, and increased risk of infection [137]. These effects are worsened by aging and comorbid conditions such as obesity and diabetes, which raise concerns about chronic PI3K inhibition in elderly humans afflicted by neurodegenerative diseases.

Even isoform-selective inhibitors display toxicological profiles. Long-term use of AS605240, a PI3Kγ inhibitor, has been linked to elevated hepatic enzymes, gastrointestinal inflammation, and rare behavioral changes. Idelalisib and duvelisib, approved for hematological malignancies, have also shown dose-dependent toxicities of long-term use such as colitis, pneumonitis, and hepatotoxicity [138]. While some of these effects are mediated through peripheral immune cells, leaky CNS microglia due to off-target PI3K activity could silence neuroinflammatory processes or change neuroinflammatory PI3K signaling.

One more consideration is adaptive resistance. Chronic inhibition of PI3K might invoke an upregulation of alternate survival pathways, including inflammatory ones, like MAPK, JAK-STAT, or NFAT, thereby diminishing drug efficacy while simultaneously modifying microglial phenotypes in ways that are hard to predict. In oncology, this has resulted in tumor escape and subsequent treatment failure [139]; a comparable phenomenon in neurodegeneration could limit cognitive benefit and worsen pathology by an incomplete suppression of neuroinflammation alongside an impaired ability to neurobiologically repair inflammation-driven damage.

Additionally, long-term inhibition could disrupt homeostatic processes governed by PI3K, including the regulation of autophagy, phagocytosis of apoptotic debris, and synapse pruning. Dampening PI3K signaling too aggressively could result in the accumulation of toxic proteins, the failure of neurovascular coupling, or a diminishment of activity-dependent plasticity. Such effects may not be obvious at first but could manifest in the context of chronic treatment, particularly in cognitively healthy people undergoing pre-emptive therapy.

Another issue that seems to be lacking is the potential consequence of developmental PI3K exposure during the critical window of gestation or adolescence. Because PI3K is a critical enzyme during CNS development, especially in early life, PI3K inhibitors could inadvertently alter brain structure and function [140]. There is currently a lack of information pertaining to teratogenic risks, or safety in pediatric populations, which complicates research designing early intervention strategies in genetically at-risk young children, such as epithelial carriers of APOE4.

The potential for psychological and behavioral side effects necessitates additional analysis. PI3K is also involved in the regulation of mood, the entrainment of circadian rhythms, and the responsiveness to stress. Its activity suppression may increase neuropsychiatric vulnerabilities to depression, anhedonia, or anxiety. These domains should be neuropsychiatrically assessed in preclinical and early-phase human model integration.

Considering these facets together indicates that safety evaluations of PI3K-targeting therapies need to be described in longitudinal, multidimensional, and stratified by isoform and patient profile. Rather than full inhibition, therapies may be enhanced by pulsatile dosing, context-aware drug delivery, or combination therapy with counteracting agents to the negative effects.

### 9.3. Cross-Talk Between PI3K Pathway and DNA Damage Response Mechanisms

The PI3K/AKT pathways are known to intersect with the DNA damage response (DDR) systems, especially with the ATM/ATR and p53 signaling pathways in post-mitotic neurons and glial cells. Chronic inflammatory conditions lead to a sustained activation of PI3K, which, in turn, has been linked to heightened oxidative stress and more rigorously damaged DNA repair pathways [85]. Moreover, H2AX phosphorylation alongside p53 stabilization, which are typically regarded as “genotoxic stress” manifestations, are controlled by mTOR and Forkhead box transcription factors (FOXO) signaling via PI3K pathways. In microglia, these changes may eventually lead to a pro-senescent state, which potentiates maladaptive immune memory and worsens neurodegeneration. In aging or chronic inflammation, sustained PI3K-Akt activity can allow microglia to survive despite accumulating DNA damage, promoting dysfunctional or senescent microglial types. Furthermore, Akt promotes the activity of DNA repair enzymes and suppresses p53-mediated apoptosis, supporting microglia survival even under genotoxic stress. The role of PI3K in the DNA damage response within the context of neuroinflammation offers a more comprehensive understanding of the intricate molecular interactions between chronic inflammation, synaptic dysfunction, and the deterioration of cognitive functions.

### 9.4. Need for Patient-Specific Molecular Profiling

An unmet need in neuroinflammation and cognitive therapeutics is a lack of precision medicine frameworks based on individual molecular profiles that could guide interventions using PI3K processes. With the diverse presentation and progression of neurodegenerative diseases, we can easily argue that a “one-size-fits-all” strategy would be unlikely to work.

Recent research has demonstrated that patients with AD, PD, multiple sclerosis, and vascular dementia exhibit distinct differences in inflammatory signatures. These differences also extend to cytokine levels and activation of glial cells, in addition to expression patterns of PI3K isoforms and associated regulatory proteins such as PTEN, SHIP1, and TREM2 [68]. Understanding these intricacies is crucial, as unappreciated nuances may lead to therapies that are poorly suited or even dangerous for the patient.

A significant challenge is the stratification of patients by their activity profiles of the PI3K pathway due to a clinically accessible biomarker gap. While CSF analysis and PET imaging offer molecular insights, they are too invasive or costly for routine use. Promising alternatives include recently developed blood-based biomarkers and exosomal proteins in addition to transcriptomics and inflammatory proteomics at the leukocyte level [141]. Combining these approaches with neuropsychological assessment and imaging biomarkers such as PET-FDG or fMRI could enable more precise phenotyping.

Beyond monograph construction, stratification based on genetic risk factors can enhance targeting precision. Causative or predisposing variants have been linked to altered PI3K signaling with increased risk of cognitive decline in PIK3CD, AKT3, and TSC1/2 [142]. These variants may be incorporated into a polygenic risk score alongside APOE4 status and environmental exposures, thereby allowing the identification of patients most likely to benefit from targeted PI3K pathway modification.

In addition, profiling patients over time is necessary. Depending on the stage of the disease, it may be protective or detrimental to inhibit PI3K. For instance, there is the possibility of reducing the synaptic loss in the early stages, but inhibiting it during the late stages may compromise glial compensation, leading to greater cognitive decline. Defining windows of opportunity and evaluating response to treatment will require longitudinal sampling of the markers at the molecular and imaging levels.

The application of artificial intelligence (AI) and machine learning provides the possibility to derive insights from multi-faceted dimensions of a patient’s record. Algorithms predicting in multi-omics, imaging, and clinical data have the ability to classify responders, fine-tune dosing schedules, and predict side effects. Validation in diverse populations is necessary to demonstrate equity and generalizability [143].

With these aspects in mind, modeling systems at the patient level, such as stem cell-derived microglia or organoid-based systems, creates a preclinical testing bridge that is translational. These systems allow for the functional evaluation of therapies that target PI3K, thus aiding personalized medicine and drug development by evaluating the toxicity outside the body.

To summarize, risks and smarter patient selection alongside rigorous monitoring and tailored routes of care will define cognitive dysfunction intervention on PI3K in the future. Building infrastructure to support this shift will necessitate collaboration between data scientists and clinicians from distinct fields such as immunology, neuroscience, and other branches, which will strongly enhance precision medicine.

## 10. Conclusions

Microglia, the immune guardians of the central nervous system, are now considered potent modulators of cognitive health due to their surveillance, synaptic remodeling, and inflammation roles. In the past decade or so, the PI3K (phosphoinositide 3-kinase) signaling pathway has drawn attention as a microglial phenotype and function master regulator. This review aims to fill in the gaps of how microglial PI3K, particularly the γ and δ isoforms, is aberrantly activated to trigger poorly regulated inflammatory response cascades along with the disruption of synaptic plasticity and progressive cognitive decline in numerous neuropathological contexts.

Cumulative evidence derived from infection studies, ischemic and traumatic brain injuries, and proteinopathies illustrates the consequence of PI3K overactivation in driving microglia to adopt an M1-dominant, neurotoxic phenotype. The described polarization is further compounded by the activation of NF-κB, Akt, mTOR, and other downstream effectors, as well as hypoxia, metabolic dysfunction, aging, and other external factors. On the other end of the spectrum, restoring balance by PI3K inhibition, particularly through isoform-selective strategies, demonstrated remarkable efficacy in preserving synapses, rescuing long-term potentiation, and ameliorating behavioral deficits in animals.

Importantly, this signaling axis intersects with pathways defined by the PI3K, MAPK, JAK-STAT, and metabolic networks, which work together to regulate an inflammatory response. The deterioration of this equilibrium not only impacts the cognitive functions of memory and executive function but also advances the pathology of AD, PD, and vascular dementia. Thus, PI3K signaling acts as both a marker and mediator of cognitive decline.

Although significant scientific advancements have been made, there are still several obstacles to clinical implementation. The role of PI3K in microglia is isoform-specific and remains unclear, particularly within human tissues. There are safety concerns associated with the long-term modulation of this pathway due to its essential role in cell survival, metabolism, and immunity. Additionally, precise risk factors still present challenges for customizing and personalizing treatment approaches for patients.

The development of brain-penetrating, microglia-targeted modulators of PI3K requires enhanced focus, coupled with tracking longitudinal safety profiles. These frameworks are critical to identifying predictive, individualized responses and reducing off-target impacts. There needs to be a multi-omics approach, further incorporating neuroimaging along with iPSC-derived microglia and cerebral organoids to target tailored windows for intervention.

Overall, focusing on the PI3K signaling pathways of microglia presents an innovative therapeutic approach to treating cognitive dysfunction resulting from neuroinflammation. Its success will hinge on a well-defined mechanistic understanding alongside personalized, stratified, translational approaches that honor the biology and unique diversities of human neural circuitry. Perhaps the integration of molecular neuroscience, systems biology, and precision medicine will initiate a transformative epoch of neuroimmunology therapies for cognitive disorders.

## 11. Future Perspective

As microglial biology transitions from the static immunological paradigm of discrete “shapes” and states to dynamic, context-sensitive, phenotypic landscapes, PI3K signaling takes on the role of a neuroimmune gateway not simply as a signaling malignancy, but as a programmable axis of control. We envision that therapies of the future would rewrite microglial processes in real time, striking them down from synapse-tearing agents to sentinels poised as circuit stabilizers instead of targeting the suppression of inflammation. This will require a move from drugs aimed at single receptors to PI3K “modular” focused that respond to the time, place, and shifts on molecular inputs.

Cell-specific delivery systems combined with AI molecular stratification and patient-derived microglial avatars create precision immunomodulation. The PI3K axis has reshaped cognitive immunology, shifting the focus from just survival and metabolism toward neuroinflammation seamlessly transitioning from blocked to reorchestrated for cognitive resilience and repair.

## 12. Methodology for Literature Review

This review was conducted according to rigorous standards for narrative synthesis of the biomedical literature. We included original scientific articles related to neurodegenerative diseases from in vivo and in vitro studies, clinical studies, and relevant review articles. We excluded purely computational methods without biological validation. In case of overlapping findings, articles with a rigorous experimental design, such as controls, genetic models, or longitudinal human data, were preferred. Discrepancies between studies were documented and critically discussed in the text. The reference list was curated to prioritize high-impact studies and those with translational relevance to cognitive dysfunction associated with microglial PI3K signaling. Emphasis was placed on including studies from the last five years (2019–2025) to reflect current advancements while also incorporating seminal works foundational to the field.

## Figures and Tables

**Figure 1 ijms-26-07212-f001:**
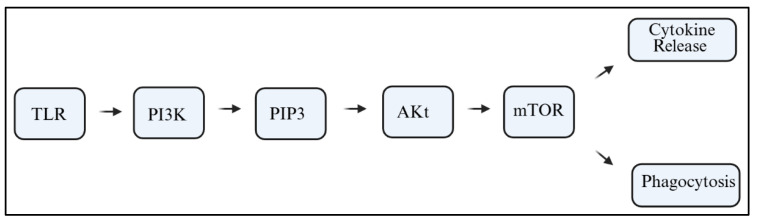
Canonical PI3K-Akt-mTOR pathway in microglia.

**Figure 2 ijms-26-07212-f002:**
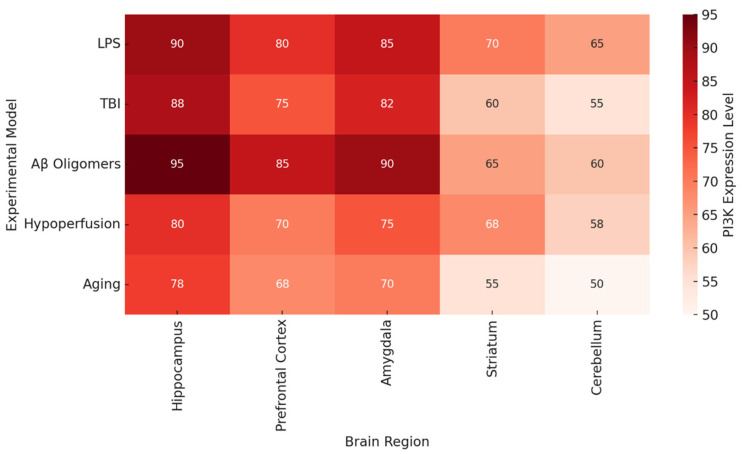
Brain region activation and PI3K expression in animal studies.

**Figure 3 ijms-26-07212-f003:**
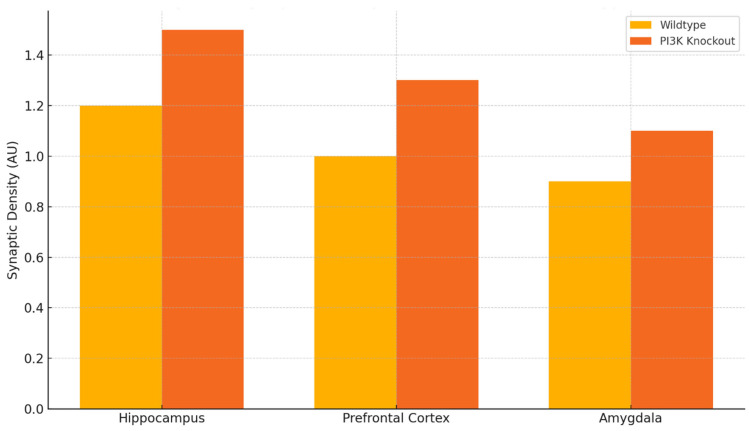
Synaptic density in PI3K knockout vs. wildtype.

**Figure 4 ijms-26-07212-f004:**
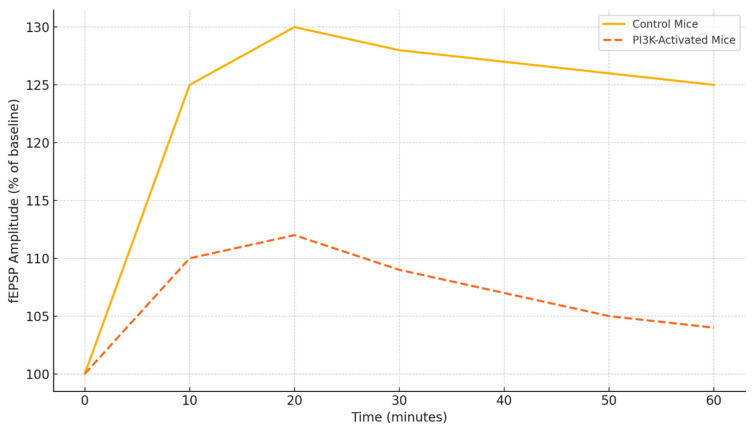
Impaired long-term potentiation (LTP) in PI3K-activated mice.

**Figure 5 ijms-26-07212-f005:**
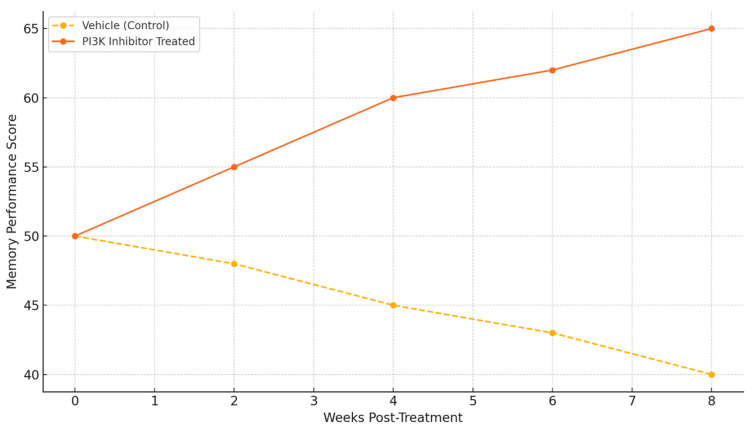
Memory scores over time with PI3K inhibitor treatment.

**Figure 6 ijms-26-07212-f006:**
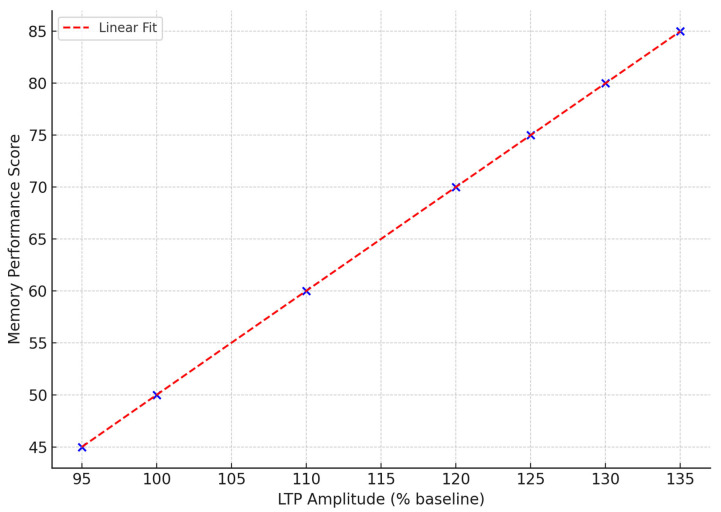
Correlation between long term potential (LTP) amplitude and memory performance.

**Figure 7 ijms-26-07212-f007:**
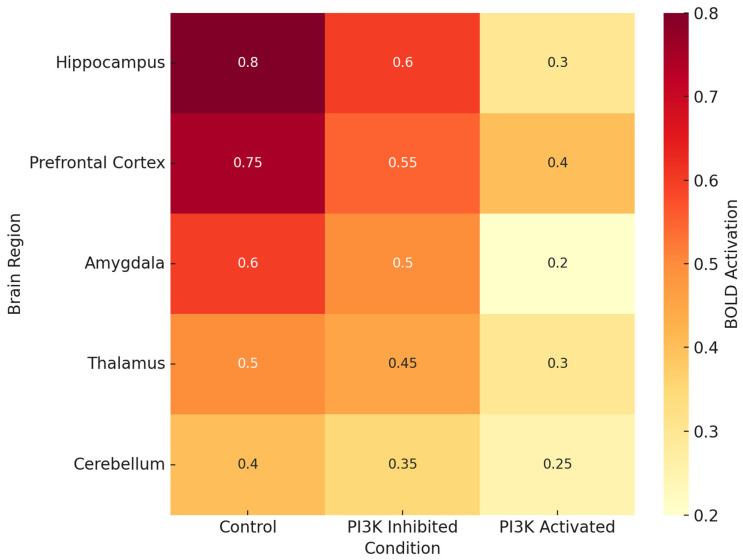
BOLD signal activation across brain regions.

**Figure 8 ijms-26-07212-f008:**
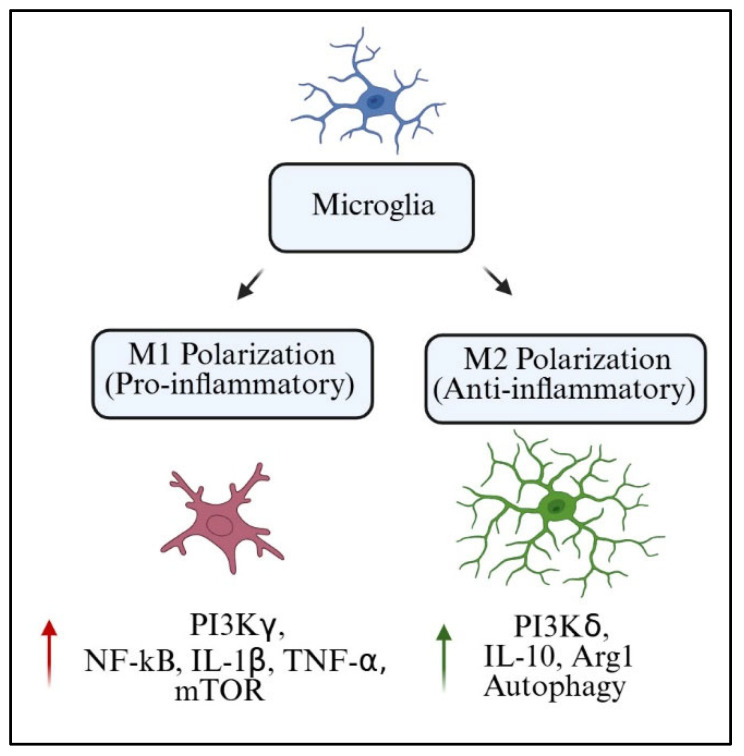
Signaling cross-talk between PI3K, NF-κB, and MAPK.

**Table 2 ijms-26-07212-t002:** Experimental models linking PI3K activation to neuroinflammation.

Model Type	PI3K Involvement	Outcome on Cognition	References
**Transgenic Mouse (APP/PS1)**	Upregulation of PI3Kγ and PI3Kδ in activated microglia	Impaired spatial memory; reversal with PI3Kγ inhibition	[35]
**LPS-Induced Inflammation**	Robust activation of PI3K-Akt-NF-κB cascade post-LPS injection	Deficits in object recognition; ameliorated by LY294002	[38]
**Traumatic Brain Injury (TBI)**	PI3K activation correlates with microglial proliferation and cytokine surge	Reduced learning performance; improved with PI3K blockade	[39]
**Chronic Cerebral Hypoperfusion**	PI3K/Akt/mTOR signaling enhances glial reactivity and cognitive impairment	Working memory impairment; restored by mTOR inhibition	[40]
**β-Amyloid Oligomer Infusion**	Local activation of PI3K in hippocampal microglia; memory deficits observed	Synaptic loss and anxiety behavior; partially rescued by PI3K inhibitor	[31]
**Aging Model (SAMP8 mice)**	Age-dependent increase in PI3K signaling and IL-1β expression	Progressive memory decline; associated with microglial PI3K hyperactivation	[32]

**Table 3 ijms-26-07212-t003:** CNS insults activating PI3K in microglia and associated outcomes.

CNS Insult Type	PI3K Activation Profile	Cognitive/Neurological Outcome	References
**Lipopolysaccharide (LPS) Injection**	Robust activation of PI3K-Akt-NF-κB; increased IL-1β and TNF-α	Impaired memory and increased anxiety behavior	[52]
**Traumatic Brain Injury (TBI)**	PI3Kγ-mediated cytokine surge and microglial proliferation	Learning and memory deficits, exacerbated with age	[60]
**Chronic Cerebral Hypoperfusion**	Sustained PI3K/mTOR activation with increased glial reactivity	Working memory and attention decline; white matter damage	[22]
**Amyloid-β Oligomers (AβO)**	Local PI3K activation in hippocampal microglia with synaptic loss	Hippocampal-dependent memory impairment and synaptic degradation	[62]
**Aging**	Baseline elevation of PI3Kγ and δ isoforms; chronic priming	Gradual cognitive decline; increased inflammatory tone	[43]
**Hypoxia-Ischemia**	Transient PI3Kα and β activation during reperfusion; delayed neuroprotection	Cognitive recovery is impaired if PI3K remains chronically active	[27,28]

**Table 4 ijms-26-07212-t004:** Summary of behavioral deficits observed in PI3K-activated models.

Experimental Model	Behavioral Deficit Observed	Linked PI3K Isoform Activity	References
**LPS-Induced Inflammation**	Impaired novel object recognition and social interaction	PI3Kγ, PI3Kδ	[52]
**Traumatic Brain Injury (TBI)**	Deficits in Morris water maze and Barnes maze navigation	PI3Kγ	[61]
**APP/PS1 AD Model**	Progressive spatial memory loss; impaired fear conditioning	PI3Kγ, PI3Kδ	[31]
**Chronic Cerebral Hypoperfusion**	Reduced attention span and working memory accuracy	PI3Kα, PI3Kβ, mTOR	[22]
**Aβ Oligomer Infusion**	Anxiety-like behavior; impaired contextual memory	PI3Kγ	[61]
**Aged Mouse (18+ months)**	Spontaneous alternation failure and delayed learning	PI3Kγ, reduced PTEN activity	[43]

**Table 5 ijms-26-07212-t005:** Pharmacological and genetic interventions targeting PI3K in cognitive studies.

Intervention Type	Primary Target	Observed Cognitive Effect	References
**Pharmacological (AS605240)**	PI3Kγ	Improved spatial memory and reduced microglial cytokine release	[105]
**Pharmacological (Idelalisib)**	PI3Kδ	Attenuated synaptic pruning; improved object recognition	[61]
**Pharmacological (LY294002)**	Class I PI3K	Reversal of LTP impairment and hippocampal inflammation	[67]
**Pharmacological (BEZ235)**	PI3K/mTOR Dual	Broad reduction in cognitive deficits with moderate toxicity	[107]
**Genetic Knockout (PI3Kγ)**	PI3Kγ gene	Rescued learning deficits and preserved synaptic density	[103]
**Genetic Knockout (PI3Kδ)**	PI3Kδ gene	Decreased inflammatory tone and improved Y-maze performance	[103]
**mTOR Inhibitor (Rapamycin)**	mTORC1 downstream of PI3K	Enhanced white matter integrity and memory recovery	[74]
**PTEN Overexpression**	PI3K regulation	Reduced microglial overactivation and memory decline	[84]
**SHIP1 Agonist (AQX-1125)**	Inhibition of PIP3 signaling	Improved executive function and reduced cytokine expression	[84]

**Table 6 ijms-26-07212-t006:** Comparative summary of cognitive readouts across methods.

Assessment Modality	Measured Outcome	PI3K-Related Observations	References
**Morris Water Maze**	Spatial learning and memory acquisition	Impaired latency and navigation in PI3K-activated mice; improved with AS605240	[59]
**Electrophysiology (LTP)**	Synaptic plasticity in hippocampal CA1	Blunted LTP induction and maintenance; restored by PI3Kγ inhibition	[68]
**fMRI BOLD Signal Analysis**	Regional activation and functional connectivity	Reduced hippocampal and PFC activation; reversed with treatment	[61]
**Y-maze Alternation Task**	Working memory and exploratory behavior	Decreased alternation accuracy in aged and inflamed mice; normalized post-treatment	[119]
**TSPO-PET Imaging**	Microglial activation and neuroinflammation	Increased tracer uptake in the hippocampus and cortex; reduced after pathway inhibition	[61]

**Table 7 ijms-26-07212-t007:** Differences in M1/M2 microglial polarization across models with PI3K involvement.

Microglial Phenotype	Canonical Markers	PI3K Involvement	References
**M1 (Pro-inflammatory)**	iNOS, IL-1β, TNF-α, CD86	PI3Kγ activation promotes M1 polarization via NF-κB and mTOR	[21]
**M2a (Anti-inflammatory)**	Arg1, IL-10, CD206, Ym1	PI3Kδ supports IL-10 production and M2a conversion under IL-4 or neurotrophic cues	[20]
**M2b (Immunoregulation)**	IL-10, TGF-β, SOCS3	Balanced PI3K activity maintains M2b homeostasis; disrupted under chronic stress	[12]
**M2c (Phagocytic)**	TREM2, CD163, MerTK	PI3K-Akt axis is essential for efferocytosis; hyperactivation impairs the resolution phase	[51]

**Table 8 ijms-26-07212-t008:** Methods for isoform-specific PI3K perturbation in experimental models and key findings.

PI3K Isoform Targeted	Perturbation Method	Cell Type/Model System	Main Experimental Findings	Key References
**PI3Kγ**	Pharmacological Inhibitor (AS605240)	Microglia (LPS-induced model)	Reduced pro-inflammatory cytokines, improved memory	[68]
**PI3Kγ**	Genetic Knockout (PI3Kγ−/−)	Microglia in Alzheimer’s disease model	Rescued synaptic plasticity and improved cognitive function	[59]
**PI3Kδ**	Pharmacological Inhibitor (Idelalisib)	Microglia (Aβ-induced neuroinflammation)	Reduced IL-1β and restored synaptic integrity	[61]
**PI3Kδ**	Genetic Knockout (PI3Kδ−/−)	Microglia in EAE (multiple sclerosis model)	Attenuated neuroinflammation and improved cognitive outcomes	[28]
**PI3Kα**	Pharmacological Inhibitor (Alpelisib)	Neurons and astrocytes (stroke models)	Reduced neuronal death; limited data on microglia	[27]
**Pan-Class I PI3K**	Pharmacological Inhibitor (LY294002)	Multiple CNS cell types (sepsis, TBI, neuroinflammation)	Reversed LTP impairment; broad anti-inflammatory effects	[30]

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
