# Peer review of "Microglia-Mediated Neuroinflammation Through Phosphatidylinositol 3-Kinase Signaling Causes Cognitive Dysfunction"

_ijms, 2025, doi:10.3390/ijms26157212_

Round 1

Reviewer 1 Report

Comments and Suggestions for Authors

Dear authors, 

First and foremost, I would like to commend you on the considerable effort invested in this review. You have undertaken an extensive and undoubtedly demanding task in compiling and synthesizing the current body of knowledge on PI3K kinase and its role in microglial activation. Your work provides a valuable and timely contribution to the field, and the comprehensive overview you have presented is greatly appreciated.

That said, I would like to offer a few suggestions that I believe could further enhance the clarity and scientific rigor of the manuscript.

In particular, I feel that the manuscript would benefit from the inclusion of additional references. The section from the text below Figure 5 to the end would be strengthened by the addition of appropriate citations, as some parts currently give the impression of having been compiled somewhat hastily.

Moreover, it would be helpful to include a brief section outlining the methodology used in preparing the review. Information such as which databases were searched, the inclusion and exclusion criteria applied, and how any discrepancies in the literature were addressed would add transparency and strengthen the overall structure of the paper.

Finally, I have included a few specific comments for your consideration:

In the Abstract
At the end of the abstract it is stated: We concludes... and the rest of the same sentence is not understandable. Please revise. 

Tables and figures should be self sufficient - please explain abbreviations used in the tables and figures. 

On which basis images 2, 3, 4, 5, 6, 7, and 8 were created? Could that be stated in the figure caption?

In the text, tables, figures - the names of the genes should be italicized. Please follow guidelines for gene nomenclature. 

The paper does not have enough references - in the following comments I have indicated where I consider this to be a problem.

In the section 2. PI3K activation in neuroinflammatory conditions

-It seems the first paragraph should contain references, but there are none. Could you please add references for those statements?

-second line: "Microglial lee center" - I am not familiar with this term. Could you please clarify? Or is it perhaps typing error?

-page 4/37 - in the 7th line there is some sort of abbreviation "pm" which is not explained. Please explain and abbreviate correctly.

-page 5/37, below Table 2, 6th line - there is an unexplained abbreviation ADC. Please explain it.

In the section 3. The PI3K Signaling Pathway: A Molecular Primer

-page 6/37 3rd line - please introduce abbreviation for CNS here, and use it in the text from now on. 

-page 7/37, 3rd line - there is repeated explanation of NF-κB. It has been abbreviated earlier in the text, therefore it is not necessary anymore. 

-page 8/37 There is a mention of the finding of Gollay and colleagues, but there is no refrence to this work. Please provide reference.

In the section 4. Microglial PI3K Activation in Response to CNS Insults

-page 9/37 - IL-6 abbreviation is again explained, but it has explanation earlier in the text. Therefore it is not necessary.

-2nd paragraph, 3rd line it says "aide" instead of "aids" I suppose?

In the section 5. Synaptic Plasticity, Memory, and PI3K Dysregulation

-page 13/37, 2nd sentence - at the end there is a quotation mark, but there is no quotation mark for beginning of the quotation. And again, there should be some reference. Please add it.

-below figure 5 - in the whole text till the section 6 there are no references at all. 

In the section 6. Therapeutic Modulation of PI3K for Cognitive Rescue

-section 6.3. has too few references.

-section 6.4. does not contain references. 

In the section 7. Cognitive Dysfunction: Mechanistic and Translational Insights

-page 21/37, paragraph starting with "Electrophysiological level" should perhaps start with "On the electrophysiological level"

-in the section 7.2 there is mention of the research by Pierce and colleagues, but there is no reference to this work, only reference 94 that is related to work of Mathys and colleagues. The rest of this section has too few references.

-text before table 6 ends with two full stops. 

-first few paragraphs of the section 7.3. miss references.

In the section 8. PI3K-Mediated Microglial Polarization and Phenotypic Shifts

-in the section 8.3. text referring to the HIF-1a misses reference as well as the next paragraph.

-section 8.4. differs in style from the rest of the sections within the section 8. 

-in the section 8.5. what does ASC stands for? Is the number 110 in the text actually a reference?

9, 10 & 11 - no remarks

In the section Abbreviations
I would rather see the abbreviation first and then its explanation.

Thank you again for your contribution, and I hope these suggestions are received in the constructive spirit in which they are offered.

Author Response

Reviewer #1 Comments:

First and foremost, I would like to commend you on the considerable effort invested in this review. You have undertaken an extensive and undoubtedly demanding task in compiling and synthesizing the current body of knowledge on PI3K kinase and its role in microglial activation. Your work provides a valuable and timely contribution to the field, and the comprehensive overview you have presented is greatly appreciated.

That said, I would like to offer a few suggestions that I believe could further enhance the clarity and scientific rigor of the manuscript.

In particular, I feel that the manuscript would benefit from the inclusion of additional references. The section from the text below Figure 5 to the end would be strengthened by the addition of appropriate citations, as some parts currently give the impression of having been compiled somewhat hastily.

Moreover, it would be helpful to include a brief section outlining the methodology used in preparing the review. Information such as which databases were searched, the inclusion and exclusion criteria applied, and how any discrepancies in the literature were addressed would add transparency and strengthen the overall structure of the paper.

Finally, I have included a few specific comments for your consideration:

Response: We thank the reviewer for the positive comments. We considered the specific suggestions of this reviewer in the revised manuscript. A point-to-point response to the reviewer is detailed below.

S. No.

Comment

Responses

1

At the end of the abstract it is stated: We concludes... and the rest of the same sentence is not understandable. Please revise.

We have revised the sentence at the end of the abstract for clarity and grammatical correctness. The corrected sentence now clearly summarizes the key conclusion of our review.

2

The section from the text below Figure 5 to the end would be strengthened by the addition of appropriate citations

We have added appropriate recent references between pages 16-18 to substantiate the discussed findings regarding synaptic plasticity, PI3K signaling, and microglial activation.

3

To include a brief section outlining the methodology used in preparing the review

We thank the reviewer for this valuable observation. A Methodology section has now been added to the manuscript on page 4, detailing the literature review strategy, databases searched, inclusion criteria, and thematic analysis approach applied in compiling and analyzing the relevant studies included in this review.

4

The paper does not have enough references - in the following comments, I have indicated where I consider this to be a problem

We have carefully revised and added appropriate recent references to support the statements presented in this section, ensuring that all key points are now backed by relevant citations.

5

In section 2. PI3K activation in neuroinflammatory conditions

-It seems the first paragraph should contain references, but there are none. Could you please add references for those statements?

We have added appropriate references to support the statements presented on page 4.

6

Second line: "Microglial lee center" - I am not familiar with this term. Could you please clarify? Or is it perhaps typing error?

We thank the reviewer for pointing out this typographical error. The phrase “microglial Lee center” has been corrected to the appropriate term, intracellular transducer, in Section 3, page 4, to ensure clarity and accuracy.

7

in the 7th line there is some sort of abbreviation "pm" which is not explained. Please explain and abbreviate correctly.

we have clarified the meaning of “pm” by explicitly stating its full form, plasma membrane, on page 4 to avoid any confusion. The abbreviation has now been defined clearly upon its first use in this section.

8

below Table 2, 6th line - there is an unexplained abbreviation ADC. Please explain it.

we have clarified the meaning of “ADC” by explicitly stating it in the text to avoid any confusion. The abbreviation has now been defined clearly upon its first use in this section, page 5.

9

please introduce abbreviation for CNS here, and use it in the text from now on

We have revised the manuscript to introduce and define the abbreviation “CNS” (central nervous system) upon its first use in the text for clarity and consistency in page 8.

10

there is repeated explanation of NF-κB. It has been abbreviated earlier in the text, therefore it is not necessary anymore

We have removed the repeated explanation of NF-κB in the later section of the manuscript to avoid redundancy in page 9

11

There is a mention of the finding of Gollay and colleagues, but there is no refrence to this work

Upon reviewing the manuscript, we found that the reference attributed to “Golay and colleagues” was incorrectly cited. We have now removed the incorrect mention of “Golay and colleagues” in page 8

12

IL-6 abbreviation is again explained, but it has explanation earlier in the text. Therefore it is not necessary.

We have removed the repeated explanation of IL-6 on page 9

13

3rd line it says "aide" instead of "aids" I suppose?

We have corrected the misuse of the word “aide” on page 9 by replacing it with aids

14

2nd sentence - at the end there is a quotation mark, but there is no quotation mark for beginning of the quotation

We have corrected the wrongly placed quotation mark on page 13

15

below figure 5 - in the whole text till the section 6 there are no references at all

We have thoroughly reviewed Section 6 and incorporated several recent, high-quality references to strengthen the discussion and support the points presented on pages 16-18.

16

paragraph starting with "Electrophysiological level" should perhaps start with "On the electrophysiological level

We have revised the phrase “electrophysiological level” on page 22 to “on the electrophysiological level.”

17

there is mention of the research by Pierce and colleagues, but there is no reference to this work

Corrected misuse of the word and corrected the sentence on page 23

18

text before table 6 ends with two full stops.

We have corrected the typo error on page 24

19

first few paragraphs of the section 7. miss references

We have added appropriate references to support the statements presented on page 22

20

section 8.3. text referring to the HIF-1a misses reference as well as the next paragraph

We have added appropriate, recent references to this section on page 27, specifically in the paragraph discussing HIF-1α and the following paragraph

21

section 8.5. what does ASC stands for? Is the number 110 in the text actually a reference?

We have clarified the abbreviation “ASC” in Section 8.5 by expanding it to “apoptosis-associated speck-like protein containing a CARD (ASC)” upon its first use on page 29. Additionally, removed the typo text mentioning 110.

22

In the section Abbreviations

I would rather see the abbreviation first and then its explanation.

We have reformatted the Abbreviations section to list abbreviations first, followed by their corresponding full forms, following the journal’s style requirements in page 32.

Reviewer 2 Report

Comments and Suggestions for Authors

It would be very wise to update references. Only 20% of all references are from the last six years (2019-2025)

Author Response

We thank the reviewer for the positive comments. Specific comments of this reviewer are addressed below.

S. No.

Comment

Responses

1

It would be very wise to update references. Only 20% of all references are from the last six years (2019-2025)

We have now incorporated multiple high-impact studies published between 2019 and 2025 to better reflect the latest advancements in microglial PI3K signaling, neuroinflammation, and cognitive dysfunction.

Reviewer 3 Report

Comments and Suggestions for Authors

The authors have written a detailed and extended review focused on microglia-mediated neuroinflammation resulting in cognitive dysfunction via the PI3K pathway. This is a unique addition within the microglial space, connecting well-studied signaling pathways and brain inflammation. The authors break down the review into 8 main sections excluding the introduction and conclusion paragraphs, beginning with PI3K activation in neuroinflammation till PI3K-mediated microglial polarization and phenotypic shifts. The authors proceed through a molecular primer of PI3K, its activation in microglia, dysregulation, cognitive consequences, M1/M2 polarization, signaling crosstalk and future directions all within the context of the CNS. In addition to the depth of information the authors provide, some sections can be expanded a bit more and others can be combined for easier reading. It would also be useful to include a short paragraph relating the importance of PI3K signaling in peripheral cancers and immune responses, mostly for context. Another consideration for the authors is to include a section on the blood-brain-barrier and the importance of this crosstalk between PI3K signaling and neuroinflammation in that space. Overall, this is a deep review on a specific signaling mode in neuroinflammation, bringing to light some important therapeutic studies and findings. Additionally, the authors should carefully go through the manuscript and fix all typos/grammatical errors for better readability. The current version contains some portions that are difficult for comprehension. Please find below some of my main comments and suggestions:

  1. The last sentence of the abstract has a typo. We “concludes” should be changed to “conclude”
  2. The first sentence of the first introductory paragraph has a typo. Please note omission of “a” in the sentence, “Cognitive impairment is associated with “a” wide range….”
  3. Sentence 4 of the second paragraph needs to be amended for grammatical flow. Firstly, sentence is too long and reads as run-on. Secondly, consider breaking the sentence into two for better coherence.
  4. Amend the first sentence of paragraph 4 for grammar.
  5. Section 2. “PI3K activation in neuroinflammatory conditions”. Beginning paragraph, especially the first couple of sentence contains multiple typos. Please amend.
  6. Second paragraph of Section 2 indicates that there are multiple isoforms of PI3K. This information was relayed in an earlier paragraph in the previous section. Please combine to avoid redundancy, unless biological significance of each isoform is being stated in the new section.
  7. How was Figure 2 generated? They y-axis is labeled as PI3K activation intensity in arbritary units? If LPS is significantly maximal on activation strength, this needs to be evident on the graph. Without statistical markers, it’s difficult to gauge the significance of each CNS insult towards PI3K activation intensity. Moreover, please define activation intensity. How is this value measured?
  8. Section 9.3 has a small paragraph with regards to PI3K pathway and DDR mechanisms. Consider expanding a bit more since a general audience may not be privy to ATM/ATR and Histone 2AX functions within the DDR context, let alone the PI3K pathway.
  9. Consider generating a summary schematic of PI3K signaling, neuroinflammation, and cognitive dysfunction, which combines the manuscript into main themes. This will incredibly help readability and comprehension.
Comments on the Quality of English Language

The language throughout the manuscript is fine in a general sense. However, there are plenty of sentence structure and syntax errors that needs to be fixed for better readability. Some paragraphs are long and can be condensed while others are shorter and can be expanded. Please amend typos throughout the manuscript.

Author Response

The authors have written a detailed and extended review focused on microglia-mediated neuroinflammation resulting in cognitive dysfunction via the PI3K pathway. This is a unique addition within the microglial space, connecting well-studied signaling pathways and brain inflammation. The authors break down the review into 8 main sections excluding the introduction and conclusion paragraphs, beginning with PI3K activation in neuroinflammation till PI3K-mediated microglial polarization and phenotypic shifts. The authors proceed through a molecular primer of PI3K, its activation in microglia, dysregulation, cognitive consequences, M1/M2 polarization, signaling crosstalk and future directions all within the context of the CNS. In addition to the depth of information the authors provide, some sections can be expanded a bit more and others can be combined for easier reading. It would also be useful to include a short paragraph relating the importance of PI3K signaling in peripheral cancers and immune responses, mostly for context. Another consideration for the authors is to include a section on the blood-brain-barrier and the importance of this crosstalk between PI3K signaling and neuroinflammation in that space. Overall, this is a deep review on a specific signaling mode in neuroinflammation, bringing to light some important therapeutic studies and findings. Additionally, the authors should carefully go through the manuscript and fix all typos/grammatical errors for better readability. The current version contains some portions that are difficult for comprehension. Please find below some of my main comments and suggestions:

We thank the reviewer for the positive comments. A point-to-point response to the reviewer were addressed below.

S. No.

Comment

Responses

1

The last sentence of the abstract has a typo. We “concludes” should be changed to “conclude”

We have fixed the typographical error in the revised document.

2

The first sentence of the first introductory paragraph has a typo. Please note omission of “a” in the sentence, “Cognitive impairment is associated with “a” wide range….”

We have amended this in the updated manuscript.

3

Sentence 4 of the second paragraph needs to be amended for grammatical flow. Firstly, sentence is too long and reads as run-on. Secondly, consider breaking the sentence into two for better coherence.

We have divided the sentence into 3 sections as recommended by the reviewer. Refer to page 2, paragraph 2 for this updated version.

4

Amend the first sentence of paragraph 4 for grammar.

The grammatical corrections of the first sentence of the fourth paragraph were addressed. Refer to page 2, paragraph 4 for this revision.

5

Section 2. “PI3K activation in neuroinflammatory conditions”. Beginning paragraph, especially the first couple of sentence contains multiple typos. Please amend.

The grammatical errors in the first paragraph of section 2 have been corrected. The revised paragraph can be found on page 4.

6

Second paragraph of Section 2 indicates that there are multiple isoforms of PI3K. This information was relayed in an earlier paragraph in the previous section. Please combine to avoid redundancy, unless biological significance of each isoform is being stated in the new section.

Thanks for highlighting the repetition of the isoforms of PI3K. These two paragraphs have now been consolidated into a single paragraph, as suggested. Please refer to page 3, paragraph 3.

7

How was Figure 2 generated? The y-axis is labeled as PI3K activation intensity in arbitrary units? If LPS is significantly maximal on activation strength, this needs to be evident on the graph. Without statistical markers, it’s difficult to gauge the significance of each CNS insult towards PI3K activation intensity. Moreover, please define activation intensity. How is this value measured?

We concur with the reviewer's argument. Since Figure 2 does not offer a statistical comparison of PI3K activation following LPS treatment across various CNS insults, the importance of the data is lessened. As a result, we have removed this figure and the related data from the updated manuscript.

8

Section 9.3 has a small paragraph with regards to PI3K pathway and DDR mechanisms. Consider expanding a bit more since a general audience may not be privy to ATM/ATR and Histone 2AX functions within the DDR context, let alone the PI3K pathway.

We have modified this paragraph to incorporate additional information regarding the role of PI3K signaling in aging and neuroinflammation. Please refer to Page 30, Paragraph 6.

9

Consider generating a summary schematic of PI3K signaling, neuroinflammation, and cognitive dysfunction, which combines the manuscript into main themes. This will incredibly help readability and comprehension.

PI3K signaling is recognized for its proinflammatory impact on microglia, which could lead to cognitive impairments. However, the influence of PI3K signaling varies depending on the isoform. While there have been notable scientific advancements, there are still numerous barriers to clinical application. The function of PI3K in microglia is specific to the isoform and remains ambiguous, especially concerning human tissues. Safety concerns are present in relation to the long-term alteration of this pathway due to its fundamental importance in cell survival, metabolism, and immune response. Furthermore, specific risk factors continue to complicate efforts to tailor and personalize treatment strategies for patients. We have incorporated these points into section 10, page 32, paragraph 4 of the updated manuscript.

Reviewer 4 Report

Comments and Suggestions for Authors

The authors have presented a well balanced review for PI3K-mTOR pathway regulations with an emphasis on isoform-specific impacts. This review would be very helpful for future research designs and should be considered for publication after the following minor revisions:

1. please provide reference for the statement "PI3Kγ and PI3Kδ are more restricted to the leukocyte lineage, which includes microglia and macrophages that infiltrate".

2. In the statement of "Golay and colleagues demonstrated that genetic...", this review failed to find a reference that involves the name Golay in the bibliography. Please consider referencing their paper after this statement.

3. It would be beneficial for readers to have a table for all the methods (inhibitors, KI/KOs, etc.) to achieve isoform-specific perturbation of PI3K activity, the cell type (if known), and the main conclusions. This would be greatly helpful to elicit future endeavors. Maybe consider merging this table with the current Table 3? I'll leave this for the authors to decide.

4. This is a major concern. To avoid confusion, Figures 3-8 should directly cite the reference where the data was generated. Were these figures copied/recreated from their original sources? Were there any modification to the presentation and/or statistic processing of the original data? These should be VERY clearly described in the main text or in the figure captions. 

Author Response

The authors have presented a well-balanced review for PI3K-mTOR pathway regulations with an emphasis on isoform-specific impacts. This review would be very helpful for future research designs and should be considered for publication after the following minor revisions:

Response: We thank the reviewer for the constructive feedback. We considered the specific suggestions of this reviewer in the revised manuscript, which are detailed below.

S. No.

Comment

Responses

1

please provide reference for the statement "PI3Kγ and PI3Kδ are more restricted to the leukocyte lineage, which includes microglia and macrophages that infiltrate

We have added appropriate references to support the statements in the respective sections.

2

In the statement of "Golay and colleagues demonstrated that genetic...", this review failed to find a reference that involves the name Golay in the bibliography

Upon reviewing the manuscript, we found that the reference attributed to “Golay and colleagues” was incorrectly cited. We have now removed the incorrect mention of “Golay and colleagues” in page 8.

3

It would be beneficial for readers to have a table for all the methods (inhibitors, KI/KOs, etc.) to achieve isoform-specific perturbation of PI3K activity, the cell type (if known), and the main conclusions

We have added a new dedicated table summarizing all reported methods on page 28.

4

This is a major concern. To avoid confusion, Figures 3-8 should directly cite the reference where the data was generated

We have carefully revised and added appropriate recent references for the similar data that we referred.